# Finite-Sample Analysis for SARSA with Linear Function Approximation

**Shaofeng Zou**
Department of Electrical Engineering
University at Buffalo, The State University of New York
Buffalo, NY 14228
szou3@buffalo.edu

**Tengyu Xu**
Department of ECE
The Ohio State University
Columbus, OH 43210
xu.3260@osu.edu

**Yingbin Liang**
Department of ECE
The Ohio State University
Columbus, OH 43210
liang.889@osu.edu

## Abstract

SARSA is an on-policy algorithm to learn a Markov decision process policy in reinforcement learning. We investigate the SARSA algorithm with linear function approximation under the non-i.i.d. data, where a single sample trajectory is available. With a Lipschitz continuous policy improvement operator that is smooth enough, SARSA has been shown to converge asymptotically [28, 23]. However, its non-asymptotic analysis is challenging and remains unsolved due to the non-i.i.d. samples and the fact that the behavior policy changes dynamically with time. In this paper, we develop a novel technique to explicitly characterize the stochastic bias of a type of stochastic approximation procedures with time-varying Markov transition kernels. Our approach enables non-asymptotic convergence analyses of this type of stochastic approximation algorithms, which may be of independent interest. Using our bias characterization technique and a gradient descent type of analysis, we provide the finite-sample analysis on the mean square error of the SARSA algorithm. We then further study a fitted SARSA algorithm, which includes the original SARSA algorithm and its variant in [28] as special cases. This fitted SARSA algorithm provides a more general framework for *iterative* on-policy fitted policy iteration, which is more memory and computationally efficient. For this fitted SARSA algorithm, we also provide its finite-sample analysis.

## 1 Introduction

SARSA, originally proposed in [31], is an on-policy reinforcement learning algorithm, which continuously updates the behavior policy towards attaining as large an accumulated reward as possible over time. Specifically, SARSA is initialized with a state and a policy. At each time instance, it takes an action based on the current policy, observes the next state, and receives a reward. Using the newly observed information, it first updates the estimate of the action-value function, and then improves the behavior policy by applying a policy improvement operator, e.g., $\epsilon$-greedy, to the estimated action-value function. Such a process is iteratively taken until it converges (see Algorithm 1 for a precise description of the SARSA algorithm).

With the tabular approach that stores the action-value function, the convergence of SARSA has been established in [33]. However, the tabular approach may not be applicable when the state space is

large or continuous. For this purpose, SARSA that incorporates parametrized function approximation is commonly used, and is more efficient and scalable. With the function approximation approach, SARSA is not guaranteed to converge in general when the $\epsilon$-greedy or softmax policy improvement operators are used [13, 10]. However, under certain conditions, its convergence can be established. For example, a variant of SARSA with linear function approximation was constructed in [28], where between two policy improvements, a temporal difference (TD) learning algorithm is applied to learn the action-value function till its convergence. The convergence of this algorithm was established in [28] using a contraction argument under the condition that the policy improvement operator is Lipschitz continuous and the Lipschitz constant is not too large. The convergence of the original SARSA algorithm under the same Lipschitz condition was later established using an O.D.E. approach in [23].

Previous studies on SARSA in [28, 23] mainly focused on the asymptotic convergence analysis, which does not suggest how fast SARSA converges and how the accuracy of the solution depends on the number of samples, i.e., sample complexity. The goal of this paper is to provide such a non-asymptotic finite-sample analysis of SARSA and to further understand how the parameters of the underlying Markov process and the algorithm affect the convergence rate. Technically, such an analysis does not follow directly from the existing finite-sample analysis for time difference (TD) learning [4, 34] and Q-learning [32], where samples are taken by a Markov process with a fixed transition kernel. The analysis of SARSA necessarily needs to deal with samples taken from a Markov decision process with a time-varying transition kernel, and in this paper, we develop novel techniques to explicitly characterize the stochastic bias for a Markov decision process with a time-varying transition kernel, which may be of independent interest.

## 1.1 Contributions

In this paper, we design a novel approach to analyze SARSA and a more general fitted SARSA algorithm, and develop the corresponding finite-sample error bounds. In particular, we consider the on-line setting where a single sample trajectory with Markovian noise is available, i.e., samples are not identical and independently distributed (i.i.d.).

**Bias characterization for time-varying Markov process.** One major challenge in our analysis is due to the fact that the estimate of the "gradient" is biased with non-i.i.d. Markovian noise. Existing studies mostly focus on the case where the samples are generated according to a Markov process with a *fixed* transition kernel, e.g., TD learning [4, 34] and Q-learning with nearest neighbors [32], so that the uniform ergodicity of the Markov process can be exploited to decouple the dependency on the Markovian noise, and then to explicitly bound the stochastic bias. For Markov processes with a *time-varying* transition kernel, such a property of uniform ergodicity does not hold in general. In this paper, we develop a novel approach to explicitly characterize the stochastic bias induced by non-i.i.d. samples generated from Markov processes with *time-varying* transition kernels. The central idea of our approach is to construct auxiliary Markov chains, which are uniformly ergodic, to approximate the dynamically changing Markov process to facilitate the analysis. Our approach can also be applied more generally to analyze stochastic approximation (SA) algorithms with time-varying Markov transition kernels, which may be of independent interest.

**Finite-sample analysis for on-policy SARSA.** For the on-policy SARSA algorithm, as the estimate of the action-value function changes with time, the behavior policy also changes. By a gradient descent type of analysis [4] and our bias characterization technique for analyzing time-varying Markov processes, we develop the finite-sample analysis for the on-policy SARSA algorithm with a continuous state space and linear function approximation. Our analysis is for the on-line case with a single sample trajectory and non-i.i.d. data. To the best of our knowledge, this is the first finite-sample analysis for this type of on-policy algorithm with time-varying behavior policy.

**Fitted SARSA algorithm.** We propose a more general on-line fitted SARSA algorithm, where between two policy improvements, a "fitted" step is taken to obtain a more accurate estimate of the action-value function of the corresponding behavior policy via multiple iterations rather than taking only a single iteration as in the original SARSA. In particular, it includes the variant of SARSA in [28] as a special case, in which each fitted step is required to converge before doing policy improvement. We provide a non-asymptotic analysis for the convergence of the proposed algorithm. Interestingly, our analysis indicates that the fitted step can stop at any time (not necessarily until convergence) without affecting the overall convergence of the fitted SARSA algorithm.

## 1.2 Related Work

**Finite-sample analysis for TD learning.** The asymptotic convergence of the TD algorithm was established in [36]. The finite-sample analysis of the TD algorithm was provided in [9, 19] under the i.i.d. setting and in [4, 34] recently under the non-i.i.d. setting, where a single sample trajectory is available. The finite sample analysis for the two-time scale methods for TD learning was also studied very recently under i.i.d. setting in [8], under non-i.i.d. setting with constant step sizes in [15], and under non-i.i.d. setting with diminishing step sizes in [38]. Differently from TD, the goal of which is to estimate the value function of a fixed policy, SARSA aims to continuously update its estimate of the action-value function to obtain an optimal policy. While samples of the TD algorithm are generated by following a *time-invariant* behavior policy, the behavior policy that generates samples in SARSA follows from an instantaneous estimate of the action-value function, which *changes over time*.

**Q-learning with function approximation.** The asymptotic convergence of Q-learning with linear function approximation was established in [23] under certain conditions. An approach based on a combination of Q-learning and kernel-based nearest neighbor regression was proposed in [32] which first discretize the entire state space, and then use the nearest neighbor regression method to estimate the action-value function. Such an approach was shown to converge, and a finite-sample analysis of the convergence rate was further provided. Q-learning algorithms in [23, 32] are off-policy algorithms, where a fixed behavior policy is used to collect samples, whereas SARSA is an on-policy algorithm with a time-varying behavior policy. Moreover, differently from the nearest neighbor approach, we consider SARSA with linear function approximation. These differences require different techniques to characterize the non-asymptotic convergence rate.

**On-policy SARSA algorithm.** SARSA was originally proposed in [31], and using the tabular approach its convergence was established in [33]. With function approximation, SARSA is not guaranteed to converge if $\epsilon$-greedy and softmax are used. With a smooth enough Lipschitz continuous policy improvement operator, the asymptotic convergence of SARSA was shown in [23, 28]. In this paper, we further develop the non-asymptotic finite-sample analysis for SARSA under the Lipschitz continuous condition.

**Fitted value/policy iteration algorithms.** The least-squares temporal difference learning (LSTD) algorithms have been extensively studied in [6, 5, 25, 20, 12, 29, 30, 35, 37] and references therein, where in each iteration a least square regression problem based on a batch data is solved. Approximate (fitted) policy iteration (API) algorithms further extend fitted value iteration with policy improvement. Several variants were studied, which adopt different objective functions, including least-squares policy iteration (LSPI) algorithms in [18, 21, 39], fitted policy iteration based on Bellman residual minimization (BRM) in [1, 11], and classification-based policy iteration algorithm in [22]. The fitted SARSA algorithm in this paper uses an *iterative* way (TD(0) algorithm) to estimate the action-value function between two policy improvements, which is more memory and computationally efficient than the batch method. Differently from [28], we do not require a convergent TD(0) run for each fitted step. For this algorithm, we provide its non-asymptotic convergence analysis.

## 2 Preliminaries

### 2.1 Markov Decision Process

Consider a general reinforcement learning setting, where an agent interacts with a stochastic environment, which is modeled as a Markov decision process (MDP). Specifically, we consider a MDP that consists of $(\mathcal{X}, \mathcal{A}, \mathsf{P}, r, \gamma)$, where $\mathcal{X}$ is a *continuous* state space $\mathcal{X} \subset \mathbb{R}^d$, and $\mathcal{A}$ is a finite action set. We further let $X_t \in \mathcal{X}$ denote the state at time $t$, and $A_t \in \mathcal{A}$ denote the action at time $t$. Then, the measure $\mathsf{P}$ defines the action dependent transition kernel for the underlying Markov chain $\{X_t\}_{t \geq 0}$: $\mathbb{P}(X_{t+1} \in U | X_t = x, A_t = a) = \int_U \mathsf{P}(dy|x, a)$, for any measurable set $U \subseteq \mathcal{X}$. The one-stage reward at time $t$ is given by $r(X_t, A_t)$, where $r : \mathcal{X} \times \mathcal{A} \to \mathbb{R}$ is the reward function, and is assumed to be uniformly bounded, i.e., $r(x, a) \in [0, r_{\max}]$, for any $(x, a) \in \mathcal{X} \times \mathcal{A}$. Finally, $\gamma$ denotes the discount factor.

A stationary policy maps a state $x \in \mathcal{X}$ to a probability distribution $\pi(\cdot|x)$ over $\mathcal{A}$, which does not depend on time. For a policy $\pi$, the corresponding value function $V^\pi : \mathcal{X} \to \mathbb{R}$ is defined as the expected total discounted reward obtained by actions executed according to

$\pi$: $V^\pi(x_0) = \mathbb{E}[\sum_{t=0}^\infty \gamma^t r(X_t, A_t)|X_0 = x_0]$. The action-value function $Q^\pi : \mathcal{X} \times \mathcal{A} \to \mathbb{R}$ is defined as $Q^\pi(x,a) = r(x,a) + \gamma \int_\mathcal{X} \mathsf{P}(dy|x,a)V^\pi(y)$. The goal is to find an optimal policy that maximizes the value function from any initial state. The optimal value function is defined as $V^*(x) = \sup_\pi V^\pi(x)$, $\forall x \in \mathcal{X}$. The optimal action-value function is defined as $Q^*(x,a) = \sup_\pi Q^\pi(x,a)$, $\forall (x,a) \in \mathcal{X} \times \mathcal{A}$. The optimal policy $\pi^*$ is then greedy with respect to $Q^*$. It can be verified that $Q^* = Q^{\pi^*}$. The Bellman operator $\mathbf{H}$ is defined as $(\mathbf{H}Q)(x,a) = r(x,a) + \gamma \int_\mathcal{X} \max_{b \in \mathcal{A}} Q(y,b)\mathsf{P}(dy|x,a)$. It is clear that $\mathbf{H}$ is contraction in the sup norm defined as $\|Q\|_{\sup} = \sup_{(x,a) \in \mathcal{X} \times \mathcal{A}} |Q(x,a)|$, and the optimal action-value function $Q^*$ is the fixed point of $\mathbf{H}$ [3].

## 2.2 Linear Function Approximation

Let $\mathcal{Q} = \{Q_\theta : \theta \in \mathbb{R}^N\}$ be a family of real-valued functions defined on $\mathcal{X} \times \mathcal{A}$. We consider the problem where any function in $\mathcal{Q}$ is a linear combination of a set of $N$ fixed functions $\phi_i : \mathcal{X} \times \mathcal{A} \to \mathbb{R}$ for $i = 1, \ldots, N$. Specifically, for $\theta \in \mathbb{R}^N$, $Q_\theta(x,a) = \sum_{i=1}^N \theta_i \phi_i(x,a) = \phi^T(x,a)\theta$. We assume that $\|\phi(x,a)\|_2 \leq 1$, $\forall(x,a) \in \mathcal{X} \times \mathcal{A}$, which can be ensured by normalizing $\{\phi_i\}_{i=1}^N$. The goal is to find a $Q_\theta$ with a compact representation in $\theta$ to approximate the optimal action-value function $Q^*$ with a continuous state space.

# 3 Finite-Sample Analysis for SARSA

## 3.1 SARSA with Linear Function Approximation

We consider a $\theta$-dependent behavior policy, which changes with time. Specifically, the behavior policy $\pi_{\theta_t}$ is given by $\Gamma(\phi^T(x,a)\theta_t)$, where $\Gamma$ is a policy improvement operator, e.g., greedy, $\epsilon$-greedy, softmax and mellowmax [2]. Suppose that $\{x_t, a_t, r_t\}_{t \geq 0}$ is a sample trajectory of states, actions and rewards obtained from the MDP following the time dependent behavior policy $\pi_{\theta_t}$ (see Algorithm 1). The projected SARSA with linear function approximation updates as follows:

$$\theta_{t+1} = \text{proj}_{2,R}(\theta_t + \alpha_t g_t(\theta_t)), \tag{1}$$

where $g_t(\theta_t) = \nabla_\theta Q_\theta(x_t, a_t)\mathbf{\Delta}_t = \phi(x_t, a_t)\mathbf{\Delta}_t$, $\mathbf{\Delta}_t$ denotes the temporal difference at time t: $\mathbf{\Delta}_t = r(x_t, a_t) + \gamma\phi^T(x_{t+1}, a_{t+1})\theta_t - \phi^T(x_t, a_t)\theta_t$, and $\text{proj}_{2,R}(\theta) := \arg\min_{\theta':\|\theta'\|_2 \leq R} \|\theta - \theta'\|_2$. In this paper, we refer to $g_t$ as "gradient", although it is not a gradient of any function.

---

**Algorithm 1** SARSA

**Initialization:**
$\theta_0, x_0, R, \phi_i$, for $i = 1, 2, ..., N$
**Method:**
$\pi_{\theta_0} \leftarrow \Gamma(\phi^T\theta_0)$
Choose $a_0$ according to $\pi_{\theta_0}$
**for** $t = 1, 2, ...$ **do**
    Observe $x_t$ and $r(x_{t-1}, a_{t-1})$
    Choose $a_t$ according to $\pi_{\theta_{t-1}}$
    $\theta_t \leftarrow \text{proj}_{2,R}(\theta_{t-1} + \alpha_{t-1}g_{t-1}(\theta_{t-1}))$
    **Policy improvement**: $\pi_{\theta_t} \leftarrow \Gamma(\phi^T\theta_t)$
**end for**

---

Here, the projection step is to control the norm of the gradient $g_t(\theta_t)$, which is a commonly used technique to control the gradient bias [4, 16, 17, 7, 26]. With a small step size $\alpha_t$ and a bounded gradient, $\theta_t$ does not change too fast. We note that [14] showed that SARSA converges to a bounded region, and thus $\theta_t$ is bounded for all $t \geq 0$. This implies that our analysis still holds without the projection step. We further note that even without exploiting the fact that $\theta_t$ is bounded, the finite-sample analysis for SARSA can still be obtained by combining our approach of analyzing the stochastic bias with an extension of the approach in [34]. However, to convey the central idea of characterizing the stochastic bias of a MDP with dynamically changing transition kernel, we focus on the projected SARSA in this paper.

We consider the following Lipschitz continuous policy improvement operator $\Gamma$ as in [28, 23]. For any $\theta \in \mathbb{R}^N$, the behavior policy $\pi_\theta = \Gamma(\phi^T \theta)$ is Lipschitz with respect to $\theta$: $\forall (x, a) \in \mathcal{X} \times \mathcal{A}$,

$$|\pi_{\theta_1}(a|x) - \pi_{\theta_2}(a|x)| \leq C\|\theta_1 - \theta_2\|_2, \tag{2}$$

where $C > 0$ is the Lipschitz constant. Further discussion about this assumption and its impact on the convergence is provided in Section 5. We further assume that for any fixed $\theta \in \mathbb{R}^N$, the Markov chain $\{X_t\}_{t \geq 0}$ induced by the behavior policy $\pi_\theta$ and the transition kernel P is uniformly ergodic with the invariant measure denoted by $\mathsf{P}_\theta$, and satisfies the following assumption.

**Assumption 1.** *There are constants $m > 0$ and $\rho \in (0, 1)$ such that*

$$\sup_{x \in \mathcal{X}} d_{TV}(\mathbb{P}(X_t \in \cdot|X_0 = x), \mathsf{P}_\theta) \leq m\rho^t, \forall t \geq 0,$$

*where $d_{TV}(P, Q)$ denotes the total-variation distance between the probability measures $P$ and $Q$.*

We denote by $\mu_\theta$ the probability measure induced by the invariant measure $\mathsf{P}_\theta$ and the behavior policy $\pi_\theta$. We assume that the $N$ base functions $\phi_i$'s are linearly independent in the Hilbert space $L^2(\mathcal{X} \times \mathcal{A}, \mu_{\theta^*})$, where $\theta^*$ is the limit point of Algorithm 1, which will be defined in the next section. For the space $L^2(\mathcal{X} \times \mathcal{A}, \mu_{\theta^*})$, two measurable functions on $\mathcal{X} \times \mathcal{A}$ are equivalent if they are identical except on a set of $\mu_{\theta^*}$-measure zero.

### 3.2 Finite-Sample Analysis

We first define $A_\theta = \mathbb{E}_\theta[\phi(X, A)(\gamma\phi^T(Y, B) - \phi^T(X, A))]$, and $b_\theta = \mathbb{E}_\theta[\phi(X, A)r(X, A)]$, where $\mathbb{E}_\theta$ denotes the expectation where $X$ follows the invariant probability measure $\mathsf{P}_\theta$, $A$ is generated by the behavior policy $\pi_\theta(A = \cdot|X)$, $Y$ is the subsequent state of $X$ following action $A$, i.e., $Y$ follows from the transition kernel $\mathsf{P}(Y \in \cdot|X, A)$, and $B$ is generated by the behavior policy $\pi_\theta(B = \cdot|Y)$. It was shown in [23] that the algorithm in (1) converges to a unique point $\theta^*$, which satisfies the following relation: $A_{\theta^*}\theta^* + b_{\theta^*} = 0$, if the Lipschitz constant $C$ is not so large that $(A_{\theta^*} + C\lambda I)$ is negative definite[1].

Let $G = r_{\max} + 2R$ and $\lambda = G|\mathcal{A}|(2 + \lceil\log_\rho \frac{1}{m}\rceil + \frac{1}{1-\rho})$. Recall in (2) that the policy $\pi_\theta$ is Lipschitz with respect to $\theta$ with Lipschitz constant $C$. We then make the following assumption [28, 23].

**Assumption 2.** *The Lipschitz constant $C$ is not so large that $(A_{\theta^*} + C\lambda I)$ is negative definite, and denote the largest eigenvalue of $\frac{1}{2}\big((A_{\theta^*} + C\lambda I) + (A_{\theta^*} + C\lambda I)^T\big)$ by $-w_s < 0$.*

The following theorems present the finite-sample bound on the convergence of SARSA with diminishing and constant step sizes.

**Theorem 1.** *Consider SARSA with linear function approximation in Algorithm 1 with $\|\theta^*\|_2 \leq R$. Consider a decaying step size $\alpha_t = \frac{1}{2w(t+1)}$ for $t \geq 0$, where $w \leq w_s$. Under Assumptions 1 and 2, we have that*

$$\mathbb{E}\|\theta_T - \theta^*\|_2^2 \leq \frac{G^2(4C|\mathcal{A}|G\tau_0^2 + (12 + 2\lambda C)\tau_0 + 1)(\log T + 1)}{4w^2 T} + \frac{2G^2(\tau_0 w + w + \rho^{-1})}{w^2 T}, \tag{3}$$

*where $\tau_0 = \min\{t \geq 0 : m\rho^t \leq \alpha_T\}$. For large $T$, $\tau_0 \sim \log T$, and hence $\mathbb{E}\|\theta_T - \theta^*\|_2^2 \leq \mathcal{O}\left(\frac{\log^3 T}{T}\right)$. Thus, to guarantee the accuracy $\mathbb{E}[\|\theta_T - \theta^*\|_2^2] \leq \delta$ for a small $\delta$, the overall sample complexity is given by $\mathcal{O}(\frac{1}{\delta}\log^3 \frac{1}{\delta})$.*

Theorem 1 indicates that SARSA has a faster convergence rate than the existing finite-sample bound for Q-learning with nearest neighbors [32].

**Theorem 2.** *Consider SARSA with linear function approximation in Algorithm 1 with $\|\theta^*\|_2 \leq R$. Under Assumptions 1 and 2 and with a constant step size $\alpha_t = \alpha_0 < \frac{1}{2w_s}$ for $t > 0$, we have that*

$$\mathbb{E}\|\theta_T - \theta^*\|_2^2 \leq e^{-2\alpha_0 w_s T}\mathbb{E}\|\theta_0 - \theta^*\|_2^2 + \frac{\alpha_0 G^2((12 + 2\lambda C)\tau_0 + 4GC|\mathcal{A}|\tau_0^2 + 8/\rho + 1)}{2w_s}, \tag{4}$$

*where $\tau_0 = \min\{t \geq 0 : m\rho^t \leq \alpha_0\}$.*

If $\alpha_0$ is small enough, and $T$ is large enough, then the algorithm converges to a small neighborhood of $\theta^*$. For example, if $\alpha_t = 1/\sqrt{T}$, the upper bound converges to zero as $T \to \infty$. The proof of this theorem is a straightforward extension of that for Theorem 1.

In order for Theorems 1 and 2 to hold, the projection radius $R$ shall be chosen such that $\|\theta^*\|_2 \leq R$. However, $\theta^*$ is unknown in advance. We next provide an upper bound on $\|\theta^*\|_2$, which can be estimated in practice [4].

**Lemma 1.** *For the projected SARSA algorithm in* (1)*, the limit point $\theta^*$ satisfies that $\|\theta^*\|_2 \leq \frac{r_{\max}}{|w_l|}$, where $w_l < 0$ is the largest eigenvalue of $\frac{1}{2}(A_{\theta^*} + A_{\theta^*}^T)$.*

### 3.3 Outline of Technical Proof of Theorem 1

The major challenge in the finite-sample analysis of SARSA lies in analyzing the stochastic bias in gradient, which are two-folds: (1) non-i.i.d. samples; and (2) dynamically changing behavior policy.

First, as per the updating rule in (1), there is a strong coupling between the sample path and $\{\theta_t\}_{t \geq 0}$, because the samples are used to compute the gradient $g_t$ and then $\theta_{t+1}$, which introduces a strong dependency between $\{\theta_t\}_{t \geq 0}$, and $\{X_t, A_t\}_{t \geq 0}$, and thus the bias in $g_t$. Moreover, differently from TD learning and Q-learning, $\theta_t$ is further used (as in the policy $\pi_{\theta_t}$) to generate the subsequent actions, which makes the dependency even stronger. Although the convergence can still be established using the O.D.E. approach [23], in order to derive a finite-sample analysis, the stochastic bias in the gradient needs to be explicitly characterized, which makes the problem challenging.

Second, as $\theta_t$ updates, the transition kernel for the state-action pair $(X_t, A_t)$ changes with time. Previous analyses, e.g., [4], rely on the facts that the behavior policy is fixed and that the underlying Markov process is uniformly ergodic, so that the Markov process reaches its stationary distribution quickly. In [28], a variant of SARSA was studied, where between two policy improvements, the behavior policy is fixed, and a TD method is used to estimate its action-value function until convergence. The behavior policy is then improved using a Lipschitz continuous policy improvement operator. In this way, for each given behavior policy, the induced Markov process can reach its stationary distribution quickly so that the analysis can be conducted. The SARSA algorithm studied in this paper does not possess these nice properties. The behavior policy of the SARSA algorithm changes at each time step, and the underlying Markov process does not necessarily reach a stationary distribution due to lack of uniform ergodicity.

To provide a finite-sample analysis, our major technical novelty lies in the design of auxiliary Markov chains, which are uniformly ergodic and , to approximate the original Markov chain induced by the SARSA algorithm, and a careful decomposition of the stochastic bias. Using such an approach, the gradient bias can be explicitly characterized. Then together with a gradient descent type of analysis, we derive the finite-sample analysis for the SARSA algorithm.

To illustrate the main idea of the proof, we provide a sketch. We note that Step 3 contains our major technical contributions of bias characterization for time-varying Markov processes.

*Proof sketch.* We first introduce some notations. For any fixed $\theta \in \mathbb{R}^N$, define $\bar{g}(\theta) = \mathbb{E}_\theta[g_t(\theta)]$, where $X_t$ follows the stationary distribution $\mathsf{P}_\theta$, and $(A_t, X_{t+1}, A_{t+1})$ are subsequent actions and states generated according to the policy $\pi_\theta$ and the transition kernel $\mathsf{P}$. Here, $\bar{g}(\theta)$ can be interpreted as the noiseless gradient at $\theta$. We then define

$$\mathbf{\Lambda}_t(\theta) = \langle \theta - \theta^*, g_t(\theta) - \bar{g}(\theta) \rangle. \tag{5}$$

Thus, $\mathbf{\Lambda}_t(\theta_t)$ measures the bias caused by using non-i.i.d. samples to estimate the gradient.

**Step 1.** Error decomposition. The error at each time step can be decomposed recursively as follows:

$$\mathbb{E}[\|\theta_{t+1} - \theta^*\|_2^2] \leq \mathbb{E}[\|\theta_t - \theta^*\|_2^2] + 2\alpha_t \mathbb{E}[\langle \theta_t - \theta^*, \bar{g}(\theta_t) - \bar{g}(\theta^*) \rangle]$$
$$+ \alpha_t^2 \mathbb{E}[\|g_t(\theta_t)\|_2^2] + 2\alpha_t \mathbb{E}[\mathbf{\Lambda}_t(\theta_t)]. \tag{6}$$

**Step 2.** Gradient descent type analysis. The first three terms in (6) mimic the analysis of the gradient descent algorithm without noise, because the accurate gradient $\bar{g}_t$ at $\theta_t$ is used.

Due to the projection step in (1), $\|g_t(\theta_t)\|_2$ is upper bounded by $G$. It can also be shown that

$$\mathbb{E}[\langle \theta_t - \theta^*, \bar{g}(\theta_t) - \bar{g}(\theta^*) \rangle] \leq (\theta_t - \theta^*)^T (A_{\theta^*} + C\lambda I)(\theta_t - \theta^*). \tag{7}$$

For a not so large $C$, i.e., $\pi_\theta$ is smooth enough with respect to $\theta$, $(A_{\theta^*} + C\lambda I)$ is negative definite. Then, we have

$$\mathbb{E}[\langle \theta_t - \theta^*, \bar{g}(\theta_t) - \bar{g}(\theta^*) \rangle] \leq -w_s \mathbb{E}[\|\theta_t - \theta^*\|_2^2]. \tag{8}$$

**Step 3.** Stochastic bias analysis. This step consists of our major technical developments. The last term in (6) is the bias caused by using a single sample path with non-i.i.d. data and time-varying behavior policy. For convenience, we rewrite $\mathbf{\Lambda}_t(\theta_t)$ as $\mathbf{\Lambda}_t(\theta_t, O_t)$, where $O_t = (X_t, A_t, X_{t+1}, A_{t+1})$. Bounding this term is challenging due to the strong dependency between $\theta_t$ and $O_t$.

We first show that $\mathbf{\Lambda}_t(\theta, O_t)$ is Lipschitz in $\theta$. Due to the projection step, $\theta_t$ changes slowly with $t$. Combining the two facts, we can show that for any $\tau > 0$,

$$\mathbf{\Lambda}_t(\theta_t, O_t) \leq \mathbf{\Lambda}_t(\theta_{t-\tau}, O_t) + (6 + \lambda C)G^2 \sum_{i=t-\tau}^{t-1} \alpha_i. \tag{9}$$

Such a step is intended to decouple the dependency between $O_t$ and $\theta_t$ by considering $O_t$ and $\theta_{t-\tau}$. If the Markov chain $\{(X_t, A_t, \theta_t)\}_{t \geq 0}$ induced by SARSA was uniformly ergodic, and satisfied Assumption 1, then for any $\theta_{t-\tau}$, $O_t$ would reach its stationary distribution quickly for large $\tau$. However, such an argument is not necessarily true, since $\theta_t$ changes with time and thus the transition kernel of the Markov chain changes with time.

Our idea is to construct an auxiliary Markov chain to assist our proof. Consider the following new Markov chain. Before time $t - \tau + 1$, the states and actions are generated according to the SARSA algorithm, but after time $t - \tau + 1$, the behavior policy is kept fixed as $\pi_{\theta_{t-\tau}}$ to generate all the subsequent actions. We then denote by $\tilde{O}_t = (\tilde{X}_t, \tilde{A}_t, \tilde{X}_{t+1}, \tilde{A}_{t+1})$ the observations of the new Markov chain at time $t$ and time $t + 1$. For this new Markov chain, for large $\tau$, $\tilde{O}_t$ reaches the stationary distribution induced by $\pi_{\theta_{t-\tau}}$ and $\mathsf{P}$. It then can be shown that

$$\mathbb{E}[\mathbf{\Lambda}_t(\theta_{t-\tau}, \tilde{O}_t)] \leq 4G^2 m \rho^{\tau-1}. \tag{10}$$

The next step is to bound the difference between the Markov chain generated by the SARSA algorithm and the auxiliary Markov chain that we construct. Since the behavior policy changes slowly, due to its Lipschitz property and the small step size $\alpha_t$, the two Markov chains should not deviate from each other too much. It can be shown that for the case with diminishing step size (similar argument can be obtained for the case with constant step size),

$$\mathbb{E}[\mathbf{\Lambda}_t(\theta_{t-\tau}, O_t)] - \mathbb{E}[\mathbf{\Lambda}_t(\theta_{t-\tau}, \tilde{O}_t)] \leq \frac{C|\mathcal{A}|G^3\tau}{w} \log \frac{t}{t-\tau}. \tag{11}$$

Combining (9), (10) and (11) yields an upper bound on $\mathbb{E}[\mathbf{\Lambda}_t(\theta_t)]$.

**Step 4.** Putting the first three steps together and recursively applying Step 1 complete the proof. $\square$

## 4  Finite-sample Analysis for Fitted SARSA Algorithm

In this section, we introduce a more general on-policy fitted SARSA algorithm (see Algorithm 2), which provides a general framework for on-policy fitted policy iteration. Specifically, after each policy improvement, we perform a "fitted" step that consists of $B$ TD(0) iterations to estimate the action-value function of the current policy. This more general fitted SARSA algorithm contains the original SARSA algorithm [31] as a special case with $B = 1$ and the algorithm in [28] as another special case with $B = \infty$ (i.e., until TD(0) converges). Moreover, the entire algorithm uses only *one single* Markov trajectory, instead of restarting from state $x_0$ after each policy improvement [28]. Differently from most existing fitted policy iteration algorithms, where a regression problem for model fitting is solved between two policy improvements, our fitted SARSA algorithm does not require a convergent TD iteration process between policy improvements. As will be shown, the on-policy fitted SARSA algorithm is guaranteed to converge for an arbitrary $B$. The overall sample complexity for this fitted algorithm will be provided.

In fact, there is no need for the number $B$ of TD iterations in the fitted step to be the same. More generally, by setting the number of TD iterations differently, we can control the estimation accuracy of

---
**Algorithm 2** General Fitted SARSA
---
**Initialization:**
$\theta_0, x_0, R, \phi_i$, for $i = 1, 2, ..., N$
**Method:**
$\pi_{\theta_0} \leftarrow \Gamma(\phi^T \theta_0)$
Choose $a_0$ according to $\pi_{\theta_0}$
**for** $t = 0, 1, 2, ...$ **do**
    **TD learning of policy** $\pi_{\theta_{tB}}$:
    **for** $j = 1, ..., B$ **do**
        Observe $x_{tB+j}$ and $r(x_{tB+j-1}, a_{tB+j-1})$
        Choose $a_{tB+j}$ according to $\pi_{\theta_{tB}}$
        $\theta_{tB+j} \leftarrow \mathrm{proj}_{2,R}(\theta_{tB+j-1} + \alpha_{tB+j-1} g_{tB+j-1}(\theta_{tB+j-1}))$
    **end for**
    **Policy improvement:** $\pi_{\theta_{(t+1)B}} \leftarrow \Gamma(\phi^T \theta_{(t+1)B})$
**end for**
---

the action-value function between policy improvements using the finite-sample bound of TD [4]. Our analysis can be extended to this general scenario in a straightforward manner, but the mathematical expressions get more involved. Thus we focus on the simple case with the same $B$ to convey the central idea.

The following theorem provides the finite-sample bound on the convergence of the fitted SARSA algorithm.

**Theorem 3.** *Consider the fitted SARSA algorithm with linear function approximation as in Algorithm 2. Suppose that Assumptions 1 and 2 hold.*

*(1) With a decaying step size $\alpha_t = \frac{1}{2tw}$ for $t \geq 1$ and $w \leq w_s$, we have that*

$$
\begin{aligned}
\mathbb{E}[\|\theta_{TB} - \theta^*\|_2^2] \\
\leq \Big( 4G^2(\tau_0 + B)w + (\log T + 1)((6 + \lambda C)G^2 \tau_0 + (6.5 + \lambda C)G^2 B \\
+ C|\mathcal{A}|G^3 \tau_0^2) + 4G^2/\rho + 0.5BG^2 \Big) \big/ (w^2 BT),
\end{aligned} \tag{12}
$$

*where $\tau_0 = \inf\{nB : m\rho^{nB} \leq \alpha_{TB}\}$. For sufficiently large $T$, $\tau_0 \sim \log T$, and hence $\mathbb{E}\|\theta_T - \theta^*\|_2^2 \leq \mathcal{O}\left(\frac{\log^3 T}{T}\right)$. For any given $B$, to guarantee the accuracy $\mathbb{E}[\|\theta_{TB} - \theta^*\|_2^2] \leq \delta$ for a small $\delta$, the overall sample complexity is given by $\mathcal{O}(\frac{1}{\delta} \log^3 \frac{1}{\delta})$.*

*(2) With a constant step size $\alpha_t = \alpha_0 < \frac{1}{2w_s B}$ for $t \geq 0$, we have that*

$$
\begin{aligned}
\mathbb{E}[\|\theta_{TB} - \theta^*\|_2^2] \\
\leq e^{-2w_s B \alpha_0 T} \|\theta_0 - \theta^*\|_2^2 + \frac{\alpha_0 (BG^2 + 2(6 + \lambda C)G^2(\tau_0 + B) + 8G^2/\rho + 2|\mathcal{A}|G^3 \tau_0^2)}{2w_s},
\end{aligned} \tag{13}
$$

*where $\tau_0 = \inf\{nB : m\rho^{nB} \leq \alpha_0\}$.*

The item (2) of Theorem 3 indicates that with a small enough constant step size and a large enough $T$, the fitted SARSA algorithm converges to a small neighborhood of $\theta^*$.

Theorem 3 further implies that the fitted step can take any number of TD iterations (not necessarily to converge) without affecting the overall convergence and sample complexity of the fitted SARSA algorithm. In particular, the comparison between the original SARSA and the fitted SARSA algorithms indicates that they have the same overall sample complexity. On the other hand, the fitted SARSA algorithm is more computationally efficient due to the following two facts: (a) with the same number of samples $n_0$, the general fitted SARSA algorithm uses a fewer number $n_0/B$ of policy improvement operators; and (b) to apply the policy improvement operator, an inner product between $\phi$ and $\theta_{tB}$ needs to be computed, the complexity of which scales linearly with the size of the action space $|\mathcal{A}|$.

# 5 Discussion of Lipschitz Continuity Assumption

In this section, we discuss the Lipschitz continuity assumption on the policy improvement operator $\Gamma$, which plays an important role in the convergence of SARSA.

Using a tabular approach that stores the action-values, the convergence of the SARSA algorithm was established in [33]. However, an example given in [13] shows that SARSA with function approximation and $\epsilon$-greedy policy improvement operator is chattering, and does not converge. Later, [14] showed that SARSA converges to a bounded region, although this region may be large, and does not diverge as Q-learning with linear function approximation. One possible explanation of this non-convergent behavior of the SARSA algorithm with $\epsilon$-greedy and softmax policy improvement operators is the discontinuity in the action selection strategies [27, 10]. More specifically, a slight change in the estimate of the action-value function may result in a big change in the behavior policy, which thus yields a completely different estimate of the action-value function.

Toward further understanding the convergence of SARSA, [10] showed that the approximate value iteration with soft-max policy improvement is guaranteed to have fixed points, which however may not be unique, and [27] later showed that for any continuous policy improvement operator, fixed points of SARSA are guaranteed to exist. Then [28] developed a convergent form of SARSA by using a Lipschitz continuous policy improvement operator, and demonstrated its convergence to the unique limit point when the Lipschitz constant is not too large. As discussed in [27], the non-convergence example in [13] does not contradict the convergence result in [28], because the example does not satisfy the Lipschitz continuity condition of the policy improvement operator, which is essential to guarantee the convergence of SARSA. In this paper, we follow this line of reasoning, and consider Lipschitz continuous policy improvement operators.

As discussed in [28], the Lipschitz constant $C$ shall be chosen not so large to ensure the convergence of the SARSA algorithm. However, to ensure exploitation, one generally prefers a large Lipschitz constant $C$ so that the agent can choose actions with higher estimated action-values. In [28], an adaptive approach to choose a policy improvement operator with a proper $C$ was proposed. It was also noted in [28] that it is possible that the convergence could be obtained with a much larger $C$ than the one suggested by Theorems 1, 2 and 3.

However, an important open problem for the SARSA algorithms with Lipschitz continuous operator (also for other algorithms with continuous action selection [10]) is that there is no theoretical performance characterization of the solutions this type of algorithms produce. It is thus of future interest to further investigate the performance of the policy generated by the SARSA algorithm with Lipschitz continuous operator.

# 6 Conclusion

In this paper, we presented the first finite-sample analysis for the SARSA algorithm with continuous state space and linear function approximation. Our analysis is applicable to the on-line case with a single sample path and non-i.i.d. data. In particular, we developed a novel technique to handle the stochastic bias for dynamically changing behavior policies, which enables non-asymptotic analysis of this type of stochastic approximation algorithms. We also presented a fitted SARSA algorithm, which provides a general framework for *iterative* on-policy fitted policy iterations. We also presented the finite-sample analysis for such a fitted SARSA algorithm.

## Acknowledgement

We would like to thank the anonymous reviewer and the Area Chair for their valuable comments. The work of T. Xu and Y. Liang was supported in partby the U.S. National Science Foundation under Grants CCF-1761506, ECCS-1818904, and CCF-1801855.

## Footnotes

[1]It can be shown that if $\phi_i$'s are linearly independent in $L^2(\mathcal{X} \times \mathcal{A}, \mu_{\theta^*})$, then $A_{\theta^*}$ is negative definite [28, 36].

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
