[Supplementary Material]

# Supplementary Materials

## A  Useful Lemmas for Proof of Theorem 1

For the SARSA algorithm, define for any $\theta \in \mathbb{R}^N$,

$$\bar{g}(\theta) = \mathbb{E}_\theta \left[ \phi(X, A) \left( r(X, A) + \gamma \phi^T(Y, B)\theta - \phi^T(X, A)\theta \right) \right]. \tag{14}$$

It can be easily verified that $\bar{g}(\theta^*) = 0$. We then define $\Lambda_t(\theta) = \langle \theta - \theta^*, g_t(\theta) - \bar{g}(\theta) \rangle$.

**Lemma 2.** *For any $\theta \in \mathbb{R}^N$ such that $\|\theta\|_2 \leq R$, $\|g_t(\theta)\|_2 \leq G$.*

*Proof.* By the definition of $g_t(\theta)$, we obtain

$$
\begin{aligned}
\|g_t(\theta)\|_2 &= \left\| \phi(x_t, a_t)(r(x_t, a_t) + \gamma\phi^T(x_{t+1}, a_{t+1})\theta - \phi^T(x_t, a_t)\theta) \right\|_2 \\
&\leq |(r(x_t, a_t) + \gamma\phi^T(x_{t+1}, a_{t+1})\theta - \phi^T(x_t, a_t)\theta| \\
&\leq r_{\max} + (1 + \gamma) \|\theta\|_2 \\
&\leq G,
\end{aligned} \tag{15}
$$

where the first two inequalities are due to the assumption that $\|\phi(x, a)\|_2 \leq 1$.  $\square$

The following lemma is useful to deal with the time-varying behavior policy.

**Lemma 3.** *For any $\theta_1$ and $\theta_2$ in $\mathbb{R}^N$,*

$$d_{TV}(\mathsf{P}_{\theta_1}, \mathsf{P}_{\theta_2}) \leq |\mathcal{A}| C \left( \lceil \log_\rho m^{-1} \rceil + \frac{1}{1 - \rho} \right) \|\theta_1 - \theta_2\|_2, \tag{16}$$

*and*

$$d_{TV}(\mu_{\theta_1}, \mu_{\theta_2}) \leq |\mathcal{A}| C \left( 1 + \lceil \log_\rho m^{-1} \rceil + \frac{1}{1 - \rho} \right) \|\theta_1 - \theta_2\|_2. \tag{17}$$

*Proof.* For $\theta_i$, $i = 1, 2$, define the transition kernels respectively as follows:

$$K_i(x, dy) = \sum_{a \in \mathcal{A}} \mathsf{P}(dy|x, a) \pi_{\theta_i}(a|x). \tag{18}$$

Following from Theorem 3.1 in [24], we obtain

$$d_{TV}(\mathsf{P}_{\theta_1}, \mathsf{P}_{\theta_2}) \leq \left( \lceil \log_\rho m^{-1} \rceil + \frac{1}{1 - \rho} \right) \|K_1 - K_2\|, \tag{19}$$

where $\|\cdot\|$ is the operator norm: $\|A\| := \sup_{\|q\|_{TV}=1} \|qA\|_{TV}$, and $\|\cdot\|_{TV}$ denotes the total-variation norm. Then, we have

$$
\begin{aligned}
\|K_1 - K_2\| &= \sup_{\|q\|_{TV}=1} \left\| \int_{\mathcal{X}} q(dx)(K_1 - K_2)(x, \cdot) \right\|_{TV} \\
&= \sup_{\|q\|_{TV}=1} \int_{\mathcal{X}} \left| \int_{\mathcal{X}} q(dx)(K_1 - K_2)(x, dy) \right| \\
&\leq \sup_{\|q\|_{TV}=1} \int_{\mathcal{X}} \left| \int_{\mathcal{X}} |q(dx)||(K_1 - K_2)(x, dy)| \right| \\
&= \sup_{\|q\|_{TV}=1} \int_{\mathcal{X}} \int_{\mathcal{X}} |q(dx)| \left| \sum_{a \in \mathcal{A}} \mathsf{P}(dy|x, a)(\pi_{\theta_1}(a|x) - \pi_{\theta_2}(a|x)) \right| \\
&\leq \sup_{\|q\|_{TV}=1} \int_{\mathcal{X}} \int_{\mathcal{X}} |q(dx)| \sum_{a \in \mathcal{A}} \mathsf{P}(dy|x, a) |\pi_{\theta_1}(a|x) - \pi_{\theta_2}(a|x)| \\
&\leq |\mathcal{A}| C \|\theta_1 - \theta_2\|_2.
\end{aligned} \tag{20}
$$

By definition, $\mu_{\theta_i}(dx, a) = \mathsf{P}_{\theta_i}(dx)\pi_{\theta_i}(a|x)$, for $i = 1, 2$. Therefore, the second result follows after a few steps of simple computations.  $\square$

**Lemma 4.** *For any $\theta \in \mathbb{R}^N$ such that $\|\theta\|_2 \leq R$,*

$$\langle \theta - \theta^*, \bar{g}(\theta) - \bar{g}(\theta^*) \rangle \leq -w_s \|\theta - \theta^*\|_2^2. \tag{21}$$

*Proof.* Let $\tilde{\theta} = \theta - \theta^*$. Denote by $d\psi_\theta = \mu_\theta(dx, a)\mathsf{P}(dy|x, a)\pi_\theta(b|y)$. By the definition of $\bar{g}$, we have

$$
\begin{aligned}
&\langle \theta - \theta^*, \bar{g}(\theta) - \bar{g}(\theta^*) \rangle \\
&= \int_{x \in \mathcal{X}} \sum_{a \in \mathcal{A}} \tilde{\theta}^T \phi(x, a) r(x, a)(\mu_\theta(dx, a) - \mu_{\theta^*}(dx, a)) \\
&\quad + \int_{x \in \mathcal{X}} \sum_{a \in \mathcal{A}} \int_{y \in \mathcal{X}} \sum_{b \in \mathcal{A}} \tilde{\theta}^T \phi(x, a)(\gamma \phi^T(y, b) - \phi^T(x, a))\left(\theta d\psi_\theta - \theta^* d\psi_{\theta^*}\right) \\
&= \int_{x \in \mathcal{X}} \sum_{a \in \mathcal{A}} \tilde{\theta}^T \phi(x, a) r(x, a)(\mu_\theta(dx, a) - \mu_{\theta^*}(dx, a)) \\
&\quad + \int_{x \in \mathcal{X}} \sum_{a \in \mathcal{A}} \int_{y \in \mathcal{X}} \sum_{b \in \mathcal{A}} \tilde{\theta}^T \phi(x, a)(\gamma \phi^T(y, b) - \phi^T(x, a))\theta(d\psi_\theta - d\psi_{\theta^*}) \\
&\quad + \int_{x \in \mathcal{X}} \sum_{a \in \mathcal{A}} \int_{y \in \mathcal{X}} \sum_{b \in \mathcal{A}} \tilde{\theta}^T \phi(x, a)(\gamma \phi^T(y, b) - \phi^T(x, a))\tilde{\theta} d\psi_{\theta^*}. 
\end{aligned} \tag{22}
$$

The first term in (22) can be bounded as follows:

$$
\begin{aligned}
&\int_{x \in \mathcal{X}} \sum_{a \in \mathcal{A}} \tilde{\theta}^T \phi(X, A) r(X, A)(\mu_\theta(dx, a) - \mu_{\theta^*}(dx, a)) \\
&\leq \|\tilde{\theta}\|_2 r_{\max} \|\mu_\theta - \mu_{\theta^*}\|_{TV} \\
&\leq \|\tilde{\theta}\|_2^2 r_{\max} |\mathcal{A}| C \left(1 + \lceil \log_\rho m^{-1} \rceil + \frac{1}{1 - \rho}\right) \\
&\leq \lambda_1 C \|\tilde{\theta}\|_2^2, 
\end{aligned} \tag{23}
$$

where the second inequality follows from Lemma 3, and $\lambda_1 = r_{\max}|\mathcal{A}| \left(2 + \lceil \log_\rho m^{-1} \rceil + \frac{1}{1-\rho}\right)$.

The second term in (22) can be bounded as follows:

$$
\begin{aligned}
&\int_{x \in \mathcal{X}} \sum_{a \in \mathcal{A}} \int_{y \in \mathcal{X}} \sum_{b \in \mathcal{A}} \tilde{\theta}^T \phi(x, a)(\gamma \phi^T(y, b) - \phi^T(x, a))\theta (d\psi_\theta - d\psi_{\theta^*}) \\
&\leq \|\tilde{\theta}\|_2 (1 + \gamma) \|\theta\|_2 \|\psi_\theta - \psi_{\theta^*}\|_{TV} \\
&\leq \|\tilde{\theta}\|_2^2 (1 + \gamma) R |\mathcal{A}| C \left(2 + \lceil \log_\rho m^{-1} \rceil + \frac{1}{1 - \rho}\right) \\
&= \lambda_2 C \|\tilde{\theta}\|_2^2, 
\end{aligned} \tag{24}
$$

where the second inequality follows from Lemma 3, and $\lambda_2 = (1 + \gamma)R |\mathcal{A}| \left(2 + \lceil \log_\rho m^{-1} \rceil + \frac{1}{1-\rho}\right)$.

Let $A_{\theta^*} = \mathbb{E}_{\theta^*}[\phi(X, A)(\gamma \phi^T(Y, B) - \phi^T(X, A))]$, which is negative definite [28, 36]. The third term in (22) is equal to $\tilde{\theta}^T A_{\theta^*} \tilde{\theta}$.

Hence,

$$\langle \theta - \theta^*, \bar{g}(\theta) - \bar{g}(\theta^*) \rangle \leq \tilde{\theta}^T (A_{\theta^*} + C(\lambda_1 + \lambda_2)I)\tilde{\theta} \leq -w_s \|\theta - \theta^*\|_2^2, \tag{25}$$

where $I$ is the identity matrix, and $-w_s$ is the largest eigenvalue of $\frac{1}{2}\big((A_{\theta^*} + C(\lambda_1 + \lambda_2)I) + (A_{\theta^*} + C(\lambda_1 + \lambda_2)I)^T\big)$. $\qquad\square$

**Lemma 5.** *For all $t \geq 0$, $\Lambda_t(\theta_t) \leq 2G^2$.*

*Proof.* The result follows from Lemma 2. Specifically,

$$\mathbf{\Lambda}_t(\theta) = \langle \theta - \theta^*, g_t(\theta) - \bar{g}(\theta) \rangle \leq \|\theta - \theta^*\|_2 \|g_t(\theta) - \bar{g}(\theta)\|_2 \leq 2R2G \leq 2G^2. \quad (26)$$

$\square$

**Lemma 6.** $|\mathbf{\Lambda}_t(\theta_1) - \mathbf{\Lambda}_t(\theta_2)| \leq (6 + \lambda C)G \|\theta_1 - \theta_2\|_2.$

*Proof.* It is clear that

$$\begin{aligned}
&|\mathbf{\Lambda}_t(\theta_1) - \mathbf{\Lambda}_t(\theta_2)| \\
&= |\langle \theta_1 - \theta^*, g_t(\theta_1) - \bar{g}(\theta_1) \rangle - \langle \theta_2 - \theta^*, g_t(\theta_2) - \bar{g}(\theta_2) \rangle| \\
&= |\langle \theta_1 - \theta^*, g_t(\theta_1) - \bar{g}(\theta_1) - (g_t(\theta_2) - \bar{g}(\theta_2)) \rangle + \langle \theta_1 - \theta^* - (\theta_2 - \theta^*), g_t(\theta_2) - \bar{g}(\theta_2) \rangle| \\
&\leq 2R \|g_t(\theta_1) - \bar{g}(\theta_1) - (g_t(\theta_2) - \bar{g}(\theta_2))\|_2 + 2G \|\theta_1 - \theta_2\|_2. \quad (27)
\end{aligned}$$

The first term in (27) can be further upper bounded as follows,

$$\|g_t(\theta_1) - \bar{g}(\theta_1) - (g_t(\theta_2) - \bar{g}(\theta_2))\|_2 \leq \|g_t(\theta_1) - g_t(\theta_2)\|_2 + \|\bar{g}(\theta_1) - \bar{g}(\theta_2))\|_2. \quad (28)$$

For the first term in (28), it follows that

$$\begin{aligned}
\|g_t(\theta_1) - g_t(\theta_2)\|_2 &= \left\| \phi(x_t, a_t) \left( \gamma(\phi^T(x_{t+1}, a_{t+1})\theta_1 - \phi^T(x_{t+1}, a_{t+1})\theta_2) - \phi^T(x_t, a_t)(\theta_1 - \theta_2) \right) \right\|_2 \\
&\leq \left| \gamma\phi^T(x_{t+1}, a_{t+1})(\theta_1 - \theta_2) \right| + \left| \phi^T(x_t, a_t)(\theta_1 - \theta_2) \right| \\
&\leq (1 + \gamma) \|\theta_1 - \theta_2\|_2. \quad (29)
\end{aligned}$$

For the second term in (28), it can be shown that

$$\begin{aligned}
&\|\bar{g}(\theta_1) - \bar{g}(\theta_2)\|_2 \\
&= \Bigg\| \int_{x \in \mathcal{X}} \sum_{a \in \mathcal{A}} \phi(x, a) r(x, a)(\mu_{\theta_1}(dx, a) - \mu_{\theta_2}(dx, a)) \\
&\quad + \int_{x \in \mathcal{X}} \sum_{a \in \mathcal{A}} \int_{y \in \mathcal{X}} \sum_{b \in \mathcal{A}} \phi(x, a)(\gamma\phi^T(y, b) - \phi^T(x, a)) \left( \theta_1 d\psi_{\theta_1} - \theta_2 d\psi_{\theta_2} \right) \Bigg\|_2 \\
&= \Bigg\| \int_{x \in \mathcal{X}} \sum_{a \in \mathcal{A}} \phi(x, a) r(x, a)(\mu_{\theta_1}(dx, a) - \mu_{\theta_2}(dx, a)) \\
&\quad + \int_{x \in \mathcal{X}} \sum_{a \in \mathcal{A}} \int_{y \in \mathcal{X}} \sum_{b \in \mathcal{A}} \phi(x, a)(\gamma\phi^T(y, b) - \phi^T(x, a))\theta(d\psi_{\theta_1} - d\psi_{\theta_2}) \\
&\quad + \int_{x \in \mathcal{X}} \sum_{a \in \mathcal{A}} \int_{y \in \mathcal{X}} \sum_{b \in \mathcal{A}} \phi(x, a)(\gamma\phi^T(y, b) - \phi^T(x, a))(\theta_1 - \theta_2)d\psi_{\theta_2} \Bigg\|_2 \\
&\overset{(a)}{\leq} \lambda_1 C \|\theta_1 - \theta_2\|_2 + \lambda_2 C \|\theta_1 - \theta_2\|_2 + (1 + \gamma) \|\theta_1 - \theta_2\|_2 \\
&\leq (\lambda C + 1 + \gamma) \|\theta_1 - \theta_2\|_2 \quad (30)
\end{aligned}$$

where (a) can be shown via similar steps to those in (23) and (24), and $\lambda = G|\mathcal{A}|(2 + \lceil \log_\rho m^{-1} \rceil + \frac{1}{1-\rho})$. This completes the proof. $\square$

We next prove the major technical lemma for proving Theorem 1.

**Lemma 7.** *Consider the case with diminishing step size. For any $\tau > 0$, and $t > \tau$,*

$$\mathbb{E}[\mathbf{\Lambda}_t(\theta_t)] \leq \frac{C|\mathcal{A}|G^3\tau}{w} \log \frac{t}{t - \tau} + 4G^2 m\rho^{\tau-1} + \frac{(6 + \lambda C)G^2}{2w} \log \frac{t}{t - \tau}. \quad (31)$$

*Proof.* **Step 1.** For any $i \geq 0$, by the update rule in (1), it is clear that

$$\begin{aligned}
\|\theta_{i+1} - \theta_i\|_2 &= \left\| \text{proj}_{2,R}(\theta_i + \alpha_i g_i(\theta_i)) - \text{proj}_{2,R}(\theta_i) \right\|_2 \\
&\leq \|\alpha_i g_i(\theta_i)\|_2 \\
&\leq \alpha_i G. \quad (32)
\end{aligned}$$

Therefore, for any $\tau \geq 0$, $\|\theta_t - \theta_{t-\tau}\|_2 \leq \sum_{i=t-\tau}^{t-1} \|\theta_{i+1} - \theta_i\|_2 \leq G \sum_{i=t-\tau}^{t-1} \alpha_i$, which, together with the Lipschitz continuous property of $\mathbf{\Lambda}_t(\theta)$ in Lemma 6, implies that $|\mathbf{\Lambda}_t(\theta_t) - \mathbf{\Lambda}_t(\theta_{t-\tau})| \leq 6G^2 \sum_{i=t-\tau}^{t-1} \alpha_i$, and thus

$$\mathbf{\Lambda}_t(\theta_t) \leq \mathbf{\Lambda}_t(\theta_{t-\tau}) + (6 + \lambda C)G^2 \sum_{i=t-\tau}^{t-1} \alpha_i. \tag{33}$$

**Step 2.** For convenience, rewrite $\mathbf{\Lambda}_t(\theta_{t-\tau}) = \mathbf{\Lambda}_t(\theta_{t-\tau}, O_t)$, where $O_t = (X_t, A_t, X_{t+1}, A_{t+1})$. Conditioning on $\theta_{t-\tau}$ and $X_{t-\tau+1}$, we construct the following Markov chain:

$$X_{t-\tau+1} \to X_{t-\tau+2} \to \tilde{X}_{t-\tau+3} \to \cdots \to \tilde{X}_t \to \tilde{X}_{t+1}, \tag{34}$$

where for $k = t - \tau + 2, \ldots, t + 1$,

$$\mathbb{P}(\tilde{X}_k \in \cdot|\theta_{t-\tau}, \tilde{X}_{k-1}) = \sum_{a \in \mathcal{A}} \pi_{\theta_{t-\tau}}(\tilde{A}_{k-1} = a|\tilde{X}_{k-1})\mathsf{P}(\tilde{X}_k \in \cdot|\tilde{A}_{k-1} = a, \tilde{X}_{k-1}), \tag{35}$$

where $\tilde{X}_{t-\tau+1} = X_{t-\tau+1}$. In short, conditioning on $\theta_{t-\tau}$ and $X_{t-\tau+1}$, this new auxiliary Markov chain is generated by repeatedly applying the same behavior policy $\pi_{\theta_{t-\tau}}$.

From the construction of the Markov chain in (34), and the assumption that for any $\theta$, the Markov chain induced by repeated applying the same policy $\pi_\theta$ is uniformly ergodic, and satisfies Assumption 1, it follows that conditioning on $(\theta_{t-\tau}, X_{t-\tau+1})$, for any $k \geq t - \tau + 1$,

$$\|\mathbb{P}(\tilde{X}_k \in \cdot|\theta_{t-\tau}, X_{t-\tau+1}) - \mathsf{P}_{\theta_{t-\tau}}\|_{TV} \leq m\rho^{k-(t-\tau+1)}. \tag{36}$$

It can be shown that

$$\mathbb{E}[\mathbf{\Lambda}_t(\theta_{t-\tau}, \tilde{O}_t)|\theta_{t-\tau}, X_{t-\tau+1}] - \mathbb{E}[\mathbf{\Lambda}_t(\theta_{t-\tau}, O_t')|\theta_{t-\tau}, X_{t-\tau+1}]$$
$$\leq 2G^2(m\rho^{\tau-1} + m\rho^\tau) \leq 4G^2m\rho^{\tau-1}. \tag{37}$$

Since for any fixed $\theta$, $\mathbb{E}[\mathbf{\Lambda}_t(\theta)] = 0$, thus $\mathbb{E}[\mathbf{\Lambda}_t(\theta_{t-\tau}, O_t')|\theta_{t-\tau}, X_{t-\tau+1}] = 0$, where $O_t'$ are independently generated by $\mathsf{P}_{\theta_{t-\tau}}$ and the policy $\pi_{\theta_{t-\tau}}$. It then follows that

$$\mathbb{E}[\mathbf{\Lambda}_t(\theta_{t-\tau}, \tilde{O}_t)] \leq 2G^2(m\rho^{\tau-1} + m\rho^\tau) \leq 4G^2m\rho^{\tau-1}. \tag{38}$$

***Step 3.*** Conditioning on $\theta_{t-\tau}$ and $X_{t-\tau+1}$, we have

$$\mathbb{E}[\mathbf{\Lambda}_t(\theta_{t-\tau}, O_t)|\theta_{t-\tau}, X_{t-\tau+1}] - \mathbb{E}[\mathbf{\Lambda}_t(\theta_{t-\tau}, \tilde{O}_t)|\theta_{t-\tau}, X_{t-\tau+1}]$$
$$\leq 2G^2\|\mathbb{P}(O_t \in \cdot|\theta_{t-\tau}, X_{t-\tau+1}) - \mathbb{P}(\tilde{O}_t \in \cdot|\theta_{t-\tau}, X_{t-\tau+1})\|_{TV} \tag{39}$$

In the following, we will develop an upper bound on the total-variation norm above, i.e., bounding the difference between the Markov chain induced by the original SARSA algorithm and the newly designed auxiliary Markov chain. We first show that

$$\|\mathbb{P}(O_t \in \cdot|\theta_{t-\tau}, X_{t-\tau+1}) - \mathbb{P}(\tilde{O}_t \in \cdot|\theta_{t-\tau}, X_{t-\tau+1})\|_{TV}$$
$$\leq \|\mathbb{P}(X_t \in \cdot|\theta_{t-\tau}, X_{t-\tau+1}) - \mathbb{P}(\tilde{X}_t \in \cdot|\theta_{t-\tau}, X_{t-\tau+1})\|_{TV}$$
$$+ C|\mathcal{A}|G \sum_{i=t-\tau}^{t-1} \alpha_i + C|\mathcal{A}|G \sum_{i=t-\tau}^{t-2} \alpha_i. \tag{40}$$

The proof of (40) can be found in Section B.

To bound the first term in (40), it first can be shown that

$$\mathbb{P}(X_t \in \cdot|\theta_{t-\tau}, X_{t-\tau+1}) = \int_{\mathcal{X}} \mathbb{P}(X_{t-1} = dx, X_t \in \cdot|\theta_{t-\tau}, X_{t-\tau+1})$$
$$= \int_{\mathcal{X}} \mathbb{P}(X_{t-1} = dx|\theta_{t-\tau}, X_{t-\tau+1})\mathbb{P}(X_t \in \cdot|\theta_{t-\tau}, X_{t-\tau+1}, X_{t-1} = x), \tag{41}$$

where

$$\mathbb{P}(X_t \in \cdot | \theta_{t-\tau}, X_{t-\tau+1}, X_{t-1} = x)$$

$$= \sum_{a \in \mathcal{A}} \mathbb{P}(X_t \in \cdot, A_{t-1} = a | \theta_{t-\tau}, X_{t-\tau+1}, X_{t-1} = x)$$

$$= \sum_{a \in \mathcal{A}} \mathsf{P}(X_t \in \cdot | x, a) \mathbb{P}(A_{t-1} = a | \theta_{t-\tau}, X_{t-\tau+1}, X_{t-1} = x)$$

$$= \sum_{a \in \mathcal{A}} \mathsf{P}(X_t \in \cdot | x, a) \mathbb{E}_{\theta_{t-2}}[\mathbb{P}(A_{t-1} = a | \theta_{t-2}, \theta_{t-\tau}, X_{t-\tau+1}, X_{t-1} = x) | \theta_{t-\tau}, X_{t-\tau+1}, X_{t-1} = x]$$

$$= \sum_{a \in \mathcal{A}} \mathsf{P}(X_t \in \cdot | x, a) \mathbb{E}_{\theta_{t-2}}[\pi_{\theta_{t-2}}(a|x) | \theta_{t-\tau}, X_{t-\tau+1}, X_{t-1} = x]. \tag{42}$$

Similarly, it can be obtained that

$$\mathbb{P}(\tilde{X}_t \in \cdot | \theta_{t-\tau}, X_{t-\tau+1})$$

$$= \int_{\mathcal{X}} \mathbb{P}(\tilde{X}_{t-1} = dx, \tilde{X}_t \in \cdot | \theta_{t-\tau}, X_{t-\tau+1})$$

$$= \int_{\mathcal{X}} \mathbb{P}(\tilde{X}_{t-1} = dx | \theta_{t-\tau}, X_{t-\tau+1}) \mathbb{P}(\tilde{X}_t \in \cdot | \theta_{t-\tau}, X_{t-\tau+1}, \tilde{X}_{t-1} = x), \tag{43}$$

where

$$\mathbb{P}(\tilde{X}_t \in \cdot | \theta_{t-\tau}, X_{t-\tau+1}, \tilde{X}_{t-1} = x)$$

$$= \sum_{a \in \mathcal{A}} \pi_{\theta_{t-\tau}}(A_{t-1} = a | \tilde{X}_{t-1} = x) \mathsf{P}(\tilde{X}_t \in \cdot | x, a). \tag{44}$$

We then bound $\|\mathbb{P}(X_t \in \cdot | \theta_{t-\tau}, X_{t-\tau+1}) - \mathbb{P}(\tilde{X}_t \in \cdot | \theta_{t-\tau}, X_{t-\tau+1})\|_{TV}$ recursively as follows:

$$\|\mathbb{P}(X_t \in \cdot | \theta_{t-\tau}, X_{t-\tau+1}) - \mathbb{P}(\tilde{X}_t \in \cdot | \theta_{t-\tau}, X_{t-\tau+1})\|_{TV}$$

$$= \frac{1}{2} \int_{x' \in \mathcal{X}} \left| \mathbb{P}(X_t = dx' | \theta_{t-\tau}, X_{t-\tau+1}) - \mathbb{P}(\tilde{X}_t = dx' | \theta_{t-\tau}, X_{t-\tau+1}) \right|$$

$$= \frac{1}{2} \int_{x' \in \mathcal{X}} \left| \int_{x \in \mathcal{X}} \mathbb{P}(X_{t-1} = dx | \theta_{t-\tau}, X_{t-\tau+1}) \mathbb{P}(X_t = dx' | \theta_{t-\tau}, X_{t-\tau+1}, X_{t-1} = x) \right.$$

$$\left. - \int_{x \in \mathcal{X}} \mathbb{P}(\tilde{X}_{t-1} = dx | \theta_{t-\tau}, X_{t-\tau+1}) \mathbb{P}(\tilde{X}_t = dx' | \theta_{t-\tau}, X_{t-\tau+1}, \tilde{X}_{t-1} = x) \right|$$

$$\leq \frac{1}{2} \int_{x' \in \mathcal{X}} \int_{x \in \mathcal{X}} \left| \mathbb{P}(X_{t-1} = dx | \theta_{t-\tau}, X_{t-\tau+1}) \mathbb{P}(X_t = dx' | \theta_{t-\tau}, X_{t-\tau+1}, X_{t-1} = x) \right.$$

$$\left. - \mathbb{P}(\tilde{X}_{t-1} = dx | \theta_{t-\tau}, X_{t-\tau+1}) \mathbb{P}(\tilde{X}_t = dx' | \theta_{t-\tau}, X_{t-\tau+1}, \tilde{X}_{t-1} = x) \right|$$

$$\leq \frac{1}{2} \int_{x' \in \mathcal{X}} \int_{x \in \mathcal{X}} \left( \left| \mathbb{P}(X_{t-1} = dx | \theta_{t-\tau}, X_{t-\tau+1}) \mathbb{P}(X_t = dx' | \theta_{t-\tau}, X_{t-\tau+1}, X_{t-1} = x) \right. \right.$$

$$\left. - \mathbb{P}(\tilde{X}_{t-1} = dx | \theta_{t-\tau}, X_{t-\tau+1}) \mathbb{P}(X_t = dx' | \theta_{t-\tau}, X_{t-\tau+1}, X_{t-1} = x) \right|$$

$$+ \left| \mathbb{P}(\tilde{X}_{t-1} = dx | \theta_{t-\tau}, X_{t-\tau+1}) \mathbb{P}(X_t = dx' | \theta_{t-\tau}, X_{t-\tau+1}, X_{t-1} = x) \right.$$

$$\left. \left. - \mathbb{P}(\tilde{X}_{t-1} = dx | \theta_{t-\tau}, X_{t-\tau+1}) \mathbb{P}(\tilde{X}_t = dx' | \theta_{t-\tau}, X_{t-\tau+1}, \tilde{X}_{t-1} = x) \right| \right)$$

$$\leq \|\mathbb{P}(X_{t-1} \in \cdot | \theta_{t-\tau}, X_{t-\tau+1}) - \mathbb{P}(\tilde{X}_{t-1} \in \cdot | \theta_{t-\tau}, X_{t-\tau+1})\|_{TV}$$

$$+ \sup_{x \in \mathcal{X}} \|\mathbb{P}(X_t \in \cdot | \theta_{t-\tau}, X_{t-\tau+1}, X_{t-1} = x) - \mathbb{P}(\tilde{X}_t \in \cdot | \theta_{t-\tau}, X_{t-\tau+1}, \tilde{X}_{t-1} = x)\|_{TV}. \tag{45}$$

The second term in (45) can be bounded using (42) and (44) as follows. For any $x \in \mathcal{X}$, it follows that

$$
\|\mathbb{P}(X_t \in \cdot|\theta_{t-\tau}, X_{t-\tau+1}, X_{t-1} = x) - \mathbb{P}(\tilde{X}_t \in \cdot|\theta_{t-\tau}, X_{t-\tau+1}, \tilde{X}_{t-1} = x)\|_{TV}
$$

$$
= \frac{1}{2} \int_{x' \in \mathcal{X}} \left| \sum_{a \in \mathcal{A}} \mathsf{P}(dx'|x,a) \mathbb{E}_{\theta_{t-2}}[\pi_{\theta_{t-2}}(a|x)|\theta_{t-\tau}, X_{t-\tau+1}, X_{t-1} = x] - \mathsf{P}(dx'|x,a)\pi_{\theta_{t-\tau}}(a|x) \right|
$$

$$
\leq \frac{1}{2} \int_{x' \in \mathcal{X}} \sum_{a \in \mathcal{A}} \mathsf{P}(dx'|x,a) \left| \mathbb{E}_{\theta_{t-2}}[\pi_{\theta_{t-2}}(a|x) - \pi_{\theta_{t-\tau}}(a|x)|\theta_{t-\tau}, X_{t-\tau+1}, X_{t-1} = x] \right|
$$

$$
\leq C|\mathcal{A}|G \sum_{i=t-\tau}^{t-3} \alpha_i, \tag{46}
$$

where the last inequality is due to the fact that

$$
\|\theta_{t-2} - \theta_{t-\tau}\|_2 \leq \sum_{i=t-\tau}^{t-3} \|\theta_{i+1} - \theta_i\|_2 \leq G \sum_{i=t-\tau}^{t-1} \alpha_i, \tag{47}
$$

and $\pi_\theta$ is Lipschitz condition in $\theta$ as in (2).

Thus, $\|\mathbb{P}(X_t \in \cdot|\theta_{t-\tau}, X_{t-\tau+1}) - \mathbb{P}(\tilde{X}_t \in \cdot|\theta_{t-\tau}, X_{t-\tau+1})\|_{TV}$ can be bounded recursively as follows:

$$
\|\mathbb{P}(X_t \in \cdot|\theta_{t-\tau}, X_{t-\tau+1}) - \mathbb{P}(\tilde{X}_t \in \cdot|\theta_{t-\tau}, X_{t-\tau+1})\|_{TV}
$$

$$
\leq \|\mathbb{P}(X_{t-1} \in \cdot|\theta_{t-\tau}, X_{t-\tau+1}) - \mathbb{P}(\tilde{X}_{t-1} \in \cdot|\theta_{t-\tau}, X_{t-\tau+1})\|_{TV} + C|\mathcal{A}|G \sum_{i=t-\tau}^{t-3} \alpha_i. \tag{48}
$$

Doing this recursively implies that

$$
\|\mathbb{P}(O_t \in \cdot|\theta_{t-\tau}, X_{t-\tau+1}) - \mathbb{P}(\tilde{O}_t \in \cdot|\theta_{t-\tau}, X_{t-\tau+1})\|_{TV}
$$

$$
\leq \|\mathbb{P}(X_{t-\tau+2} \in \cdot|\theta_{t-\tau}, X_{t-\tau+1}) - \mathbb{P}(\tilde{X}_{t-\tau+2} \in \cdot|\theta_{t-\tau}, X_{t-\tau+1})\|_{TV} + C|\mathcal{A}|G \sum_{j=t-\tau}^{t-1} \sum_{i=t-\tau}^{j} \alpha_i
$$

$$
= C|\mathcal{A}|G \sum_{j=t-\tau}^{t-1} \sum_{i=t-\tau}^{j} \alpha_i
$$

$$
\leq \frac{C|\mathcal{A}|G\tau}{2w} \log \frac{t}{t-\tau}. \tag{49}
$$

Combining Steps 1, 2 and 3 completes the proof. $\qquad\square$

# B  Proof of Equation (40)

The total variation in (39) can be written as follows:

$$
\|\mathbb{P}(O_t \in \cdot|\theta_{t-\tau}, X_{t-\tau+1}) - \mathbb{P}(\tilde{O}_t \in \cdot|\theta_{t-\tau}, X_{t-\tau+1})\|_{TV}
$$

$$
= \frac{1}{2} \int_{\mathcal{X}} \sum_{\mathcal{A}} \int_{\mathcal{X}} \sum_{\mathcal{A}} \left| \mathbb{P}(X_t = dx, A_t = a, X_{t+1} = dx', A_{t+1} = a'|\theta_{t-\tau}, X_{t-\tau+1}) \right.
$$

$$
\left. - \mathbb{P}(\tilde{X}_t = dx, \tilde{A}_t = a, \tilde{X}_{t+1} = dx', \tilde{A}_{t+1} = a'|\theta_{t-\tau}, X_{t-\tau+1}) \right|. \tag{50}
$$

Here, the first term in (50) can be written as

$$\mathbb{P}(X_t = dx, A_t = a, X_{t+1} = dx', A_{t+1} = a'|\theta_{t-\tau}, X_{t-\tau+1})$$

$$= \int_{\substack{z_{t-1}\in\mathbb{R}^N \\ z_t\in\mathbb{R}^N}} \mathbb{P}(X_t = dx, A_t = a, X_{t+1} = dx', A_{t+1} = a', \theta_{t-1} = dz_{t-1}, \theta_t = dz_t|\theta_{t-\tau}, X_{t-\tau+1})$$

$$= \int_{\mathbb{R}^N}\int_{\mathbb{R}^N} \mathbb{P}(X_t = dx|\theta_{t-\tau}, X_{t-\tau+1})\mathbb{P}(\theta_{t-1} = dz_{t-1}|\theta_{t-\tau}, X_{t-\tau+1}, X_t = x)$$

$$\times \pi_{z_{t-1}}(a|x)\mathbb{P}(\theta_t = dz_t|\theta_{t-\tau}, X_{t-\tau+1}, X_t = x, A_t = a, \theta_{t-1} = z_{t-1})\mathsf{P}(dx'|x, a)\pi_{z_t}(a'|x'). \tag{51}$$

By the definition of the newly designed auxiliary Markov chain, the second term in (50) can be written as

$$\mathbb{P}(\tilde{X}_t = dx, \tilde{A}_t = a, \tilde{X}_{t+1} = dx', \tilde{A}_{t+1} = a'|\theta_{t-\tau}, X_{t-\tau+1})$$

$$= \mathbb{P}(\tilde{X}_t = dx|\theta_{t-\tau}, X_{t-\tau+1})\pi_{\theta_{t-\tau}}(a|x)\mathsf{P}(dx'|x, a)\pi_{\theta_{t-\tau}}(a'|x')$$

$$= \int_{\substack{z_{t-1}\in\mathbb{R}^N \\ z_t\in\mathbb{R}^N}} \mathbb{P}(\tilde{X}_t = dx|\theta_{t-\tau}, X_{t-\tau+1})\pi_{\theta_{t-\tau}}(a|x)\mathsf{P}(dx'|x, a)\pi_{\theta_{t-\tau}}(a'|x')$$

$$\times \mathbb{P}(\theta_{t-1} = dz_{t-1}|\theta_{t-\tau}, X_{t-\tau+1}, X_t = x)\mathbb{P}(\theta_t = dz_t|\theta_{t-\tau}, X_{t-\tau+1}, X_t = x, A_t = a, \theta_{t-1} = z_{t-1}) \tag{52}$$

Thus, by plugging (51) and (52) into (50), (50) can be further bounded as follows:

$$\|\mathbb{P}(O_t \in \cdot|\theta_{t-\tau}, X_{t-\tau+1}) - \mathbb{P}(\tilde{O}_t \in \cdot|\theta_{t-\tau}, X_{t-\tau+1})\|_{TV}$$

$$= \frac{1}{2}\int_{\mathcal{X}}\sum_{\mathcal{A}}\int_{\mathcal{X}}\sum_{\mathcal{A}}\left|\int_{\mathbb{R}^N}\int_{\mathbb{R}^N}\left(\mathbb{P}(X_t = dx|\theta_{t-\tau}, X_{t-\tau+1})\mathbb{P}(\theta_{t-1} = dz_{t-1}|\theta_{t-\tau}, X_{t-\tau+1}, X_t = x)\right.\right.$$

$$\times \pi_{z_{t-1}}(a|x)\mathbb{P}(\theta_t = dz_t|\theta_{t-\tau}, X_{t-\tau+1}, X_t = x, A_t = a, \theta_{t-1} = z_{t-1})\mathsf{P}(dx'|x, a)\pi_{z_t}(a'|x')$$

$$- \mathbb{P}(\tilde{X}_t = dx|\theta_{t-\tau}, X_{t-\tau+1})\pi_{\theta_{t-\tau}}(a|x)\mathsf{P}(dx'|x, a)\pi_{\theta_{t-\tau}}(a'|x')$$

$$\left.\left.\times \mathbb{P}(\theta_{t-1} = dz_{t-1}|\theta_{t-\tau}, X_{t-\tau+1}, X_t = x)\mathbb{P}(\theta_t = dz_t|\theta_{t-\tau}, X_{t-\tau+1}, X_t = x, A_t = a, \theta_{t-1} = z_{t-1})\right)\right|$$

$$\leq \frac{1}{2}\int_{\mathcal{X}}\sum_{\mathcal{A}}\int_{\mathcal{X}}\sum_{\mathcal{A}}\int_{\mathbb{R}^N}\int_{\mathbb{R}^N}\left|\left(\mathbb{P}(X_t = dx|\theta_{t-\tau}, X_{t-\tau+1})\mathbb{P}(\theta_{t-1} = dz_{t-1}|\theta_{t-\tau}, X_{t-\tau+1}, X_t = x)\right.\right.$$

$$\times \pi_{z_{t-1}}(a|x)\mathbb{P}(\theta_t = dz_t|\theta_{t-\tau}, X_{t-\tau+1}, X_t = x, A_t = a, \theta_{t-1} = z_{t-1})\mathsf{P}(dx'|x, a)\pi_{z_t}(a'|x')$$

$$- \mathbb{P}(\tilde{X}_t = dx|\theta_{t-\tau}, X_{t-\tau+1})\pi_{\theta_{t-\tau}}(a|x)\mathsf{P}(dx'|x, a)\pi_{\theta_{t-\tau}}(a'|x')$$

$$\left.\left.\times \mathbb{P}(\theta_{t-1} = dz_{t-1}|\theta_{t-\tau}, X_{t-\tau+1}, X_t = x)\mathbb{P}(\theta_t = dz_t|\theta_{t-\tau}, X_{t-\tau+1}, X_t = x, A_t = a, \theta_{t-1} = z_{t-1})\right)\right|. \tag{53}$$

With a slight abuse of notations, let

$$M_1 = \mathbb{P}(X_t = dx|\theta_{t-\tau}, X_{t-\tau+1})\mathbb{P}(\theta_{t-1} = dz_{t-1}|\theta_{t-\tau}, X_{t-\tau+1}, X_t = x)$$

$$\times \pi_{z_{t-1}}(a|x)\mathbb{P}(\theta_t = dz_t|\theta_{t-\tau}, X_{t-\tau+1}, X_t = x, A_t = a, \theta_{t-1} = z_{t-1})\mathsf{P}(dx'|x, a)\pi_{z_t}(a'|x'),$$

$$M_2 = \mathbb{P}(\tilde{X}_t = dx|\theta_{t-\tau}, X_{t-\tau+1})\mathbb{P}(\theta_{t-1} = dz_{t-1}|\theta_{t-\tau}, X_{t-\tau+1}, X_t = x)$$

$$\times \mathbb{P}(\theta_t = dz_t|\theta_{t-\tau}, X_{t-\tau+1}, X_t = x, A_t = a, \theta_{t-1} = z_{t-1})\pi_{\theta_{t-\tau}}(a|x)\mathsf{P}(dx'|x, a)\pi_{\theta_{t-\tau}}(a'|x'). \tag{54}$$

Thus, (53) can be written as

$$\|\mathbb{P}(O_t \in \cdot|\theta_{t-\tau}, X_{t-\tau+1}) - \mathbb{P}(\tilde{O}_t \in \cdot|\theta_{t-\tau}, X_{t-\tau+1})\|_{TV}$$

$$\leq \frac{1}{2}\int_{\mathcal{X}}\sum_{\mathcal{A}}\int_{\mathcal{X}}\sum_{\mathcal{A}}\int_{\mathbb{R}^N}\int_{\mathbb{R}^N}|M_1 - M_2|, \tag{55}$$

This can be further bounded as follows

$$\|\mathbb{P}(O_t \in \cdot|\theta_{t-\tau}, X_{t-\tau+1}) - \mathbb{P}(\tilde{O}_t \in \cdot|\theta_{t-\tau}, X_{t-\tau+1})\|_{TV}$$

$$\leq \frac{1}{2} \int_{\mathcal{X}} \sum_{\mathcal{A}} \int_{\mathcal{X}} \sum_{\mathcal{A}} \int_{\mathbb{R}^N} \int_{\mathbb{R}^N} |M_1 - M_3 + M_3 - M_2|$$

$$\leq \frac{1}{2} \int_{\mathcal{X}} \sum_{\mathcal{A}} \int_{\mathcal{X}} \sum_{\mathcal{A}} \int_{\mathbb{R}^N} \int_{\mathbb{R}^N} |M_1 - M_3| + |M_3 - M_2|, \tag{56}$$

where

$$M_3 = \mathbb{P}(\tilde{X}_t = dx|\theta_{t-\tau}, X_{t-\tau+1})\mathbb{P}(\theta_t = dz_t|\theta_{t-\tau}, X_{t-\tau+1}, X_t = x, A_t = a, \theta_{t-1} = z_{t-1})$$
$$\times \mathbb{P}(\theta_{t-1} = dz_{t-1}|\theta_{t-\tau}, X_{t-\tau+1}, X_t = x)\pi_{\theta_{t-\tau}}(a|x)\mathsf{P}(dx'|x, a)\pi_{z_t}(a'|x'). \tag{57}$$

We first consider the second term in (56):

$$\frac{1}{2} \int_{\mathcal{X}} \sum_{\mathcal{A}} \int_{\mathcal{X}} \sum_{\mathcal{A}} \int_{\mathbb{R}^N} \int_{\mathbb{R}^N} |M_3 - M_2|$$

$$= \frac{1}{2} \int_{\mathcal{X}} \sum_{\mathcal{A}} \int_{\mathcal{X}} \sum_{\mathcal{A}} \int_{\mathbb{R}^N} \int_{\mathbb{R}^N} \Big| \mathbb{P}(\tilde{X}_t = dx|\theta_{t-\tau}, X_{t-\tau+1})\pi_{\theta_{t-\tau}}(a|x)\mathsf{P}(dx'|x, a)\pi_{z_t}(a'|x')$$

$$\times \mathbb{P}(\theta_{t-1} = dz_{t-1}|\theta_{t-\tau}, X_{t-\tau+1}, X_t = x)\mathbb{P}(\theta_t = dz_t|\theta_{t-\tau}, X_{t-\tau+1}, X_t = x, A_t = a, \theta_{t-1} = z_{t-1})$$

$$- \mathbb{P}(\tilde{X}_t = dx|\theta_{t-\tau}, X_{t-\tau+1})\pi_{\theta_{t-\tau}}(a|x)\mathsf{P}(dx'|x, a)\pi_{\theta_{t-\tau}}(a'|x')$$

$$\times \mathbb{P}(\theta_{t-1} = dz_{t-1}|\theta_{t-\tau}, X_{t-\tau+1}, X_t = x)\mathbb{P}(\theta_t = dz_t|\theta_{t-\tau}, X_{t-\tau+1}, X_t = x, A_t = a, \theta_{t-1} = z_{t-1})\Big|$$

$$= \frac{1}{2} \int_{\mathcal{X}} \sum_{\mathcal{A}} \int_{\mathcal{X}} \sum_{\mathcal{A}} \int_{\mathbb{R}^N} \int_{\mathbb{R}^N} \mathbb{P}(\tilde{X}_t = dx|\theta_{t-\tau}, X_{t-\tau+1})\pi_{\theta_{t-\tau}}(a|x)\mathsf{P}(dx'|x, a)$$

$$\times \mathbb{P}(\theta_{t-1} = dz_{t-1}|\theta_{t-\tau}, X_{t-\tau+1}, X_t = x)\mathbb{P}(\theta_t = dz_t|\theta_{t-\tau}, X_{t-\tau+1}, X_t = x, A_t = a, \theta_{t-1} = z_{t-1})$$

$$\times \Big|\pi_{z_t}(a'|x') - \pi_{\theta_{t-\tau}}(a'|x')\Big|$$

$$= \int_{\mathcal{X}} \sum_{\mathcal{A}} \int_{\mathcal{X}} \int_{\mathbb{R}^N} \int_{\mathbb{R}^N} \mathbb{P}(\tilde{X}_t = dx|\theta_{t-\tau}, X_{t-\tau+1})\pi_{\theta_{t-\tau}}(a|x)\mathsf{P}(dx'|x, a)$$

$$\times \mathbb{P}(\theta_{t-1} = dz_{t-1}|\theta_{t-\tau}, X_{t-\tau+1}, X_t = x)\mathbb{P}(\theta_t = dz_t|\theta_{t-\tau}, X_{t-\tau+1}, X_t = x, A_t = a, \theta_{t-1} = z_{t-1})$$

$$\times \|\pi_{z_t}(\cdot|x') - \pi_{\theta_{t-\tau}}(\cdot|x')\|_{TV}$$

$$\leq \int_{\mathcal{X}} \sum_{\mathcal{A}} \int_{\mathcal{X}} \int_{\mathbb{R}^N} \int_{\mathbb{R}^N} \mathbb{P}(\tilde{X}_t = dx|\theta_{t-\tau}, X_{t-\tau+1})\pi_{\theta_{t-\tau}}(a|x)\mathsf{P}(dx'|x, a)$$

$$\times \mathbb{P}(\theta_{t-1} = dz_{t-1}|\theta_{t-\tau}, X_{t-\tau+1}, X_t = x)\mathbb{P}(\theta_t = dz_t|\theta_{t-\tau}, X_{t-\tau+1}, X_t = x, A_t = a, \theta_{t-1} = z_{t-1})$$

$$\times \sup_{x' \in \mathcal{X}} \|\pi_{z_t}(\cdot|x') - \pi_{\theta_{t-\tau}}(\cdot|x')\|_{TV} \tag{58}$$

Recalling that

$$\|\theta_t - \theta_{t-\tau}\|_2 \leq \sum_{i=t-\tau}^{t-1} \|\theta_{i+1} - \theta_i\|_2 \leq G \sum_{i=t-\tau}^{t-1} \alpha_i, \tag{59}$$

and by the Lipschitz condition in (2), it follows that for any $x' \in \mathcal{X}$,

$$\|\pi_{\theta_t}(\cdot|x') - \pi_{\theta_{t-\tau}}(\cdot|x')\|_{TV} \leq C|\mathcal{A}|G \sum_{i=t-\tau}^{t-1} \alpha_i. \tag{60}$$

Thus, for any $z_t$ such that $z_t dz_t$ has non-zero measure, i.e., $\theta_{t-1} = z_{t-1}$ satisfies (59), it can be shown that

$$\sup_{x' \in \mathcal{X}} \|\pi_{z_t}(\cdot|x') - \pi_{\theta_{t-\tau}}(\cdot|x')\|_{TV} \leq C|\mathcal{A}|G \sum_{i=t-\tau}^{t-1} \alpha_i, \tag{61}$$

which further implies that (58) can be further bounded by $C|\mathcal{A}|G\sum_{i=t-\tau}^{t-1}\alpha_i$.

We then consider the first term in (56):

$$\frac{1}{2}\int_{\mathcal{X}}\sum_{\mathcal{A}}\int_{\mathcal{X}}\sum_{\mathcal{A}}\int_{\mathbb{R}^N}\int_{\mathbb{R}^N}|M_1-M_3|$$

$$=\frac{1}{2}\int_{\mathcal{X}}\sum_{\mathcal{A}}\int_{\mathcal{X}}\sum_{\mathcal{A}}\int_{\mathbb{R}^N}\int_{\mathbb{R}^N}\Big|\mathbb{P}(X_t=dx|\theta_{t-\tau},X_{t-\tau+1})\mathbb{P}(\theta_{t-1}=dz_{t-1}|\theta_{t-\tau},X_{t-\tau+1},X_t=x)$$

$$\times\pi_{z_{t-1}}(a|x)\mathbb{P}(\theta_t=dz_t|\theta_{t-\tau},X_{t-\tau+1},X_t=x,A_t=a,\theta_{t-1}=z_{t-1})\mathsf{P}(dx'|x,a)\pi_{z_t}(a'|x')$$

$$-\mathbb{P}(\tilde{X}_t=dx|\theta_{t-\tau},X_{t-\tau+1})\mathbb{P}(\theta_{t-1}=dz_{t-1}|\theta_{t-\tau},X_{t-\tau+1},X_t=x)$$

$$\times\pi_{\theta_{t-\tau}}(a|x)\mathsf{P}(dx'|x,a)\pi_{z_t}(a'|x')\mathbb{P}(\theta_t=dz_t|\theta_{t-\tau},X_{t-\tau+1},X_t=x,A_t=a,\theta_{t-1}=z_{t-1})\Big|$$

$$=\frac{1}{2}\int_{\mathcal{X}}\sum_{\mathcal{A}}\int_{\mathcal{X}}\sum_{\mathcal{A}}\int_{\mathbb{R}^N}\int_{\mathbb{R}^N}\pi_{z_t}(a'|x')\mathsf{P}(dx'|x,a)\mathbb{P}(\theta_{t-1}=dz_{t-1}|\theta_{t-\tau},X_{t-\tau+1},X_t=x)$$

$$\times\mathbb{P}(\theta_t=dz_t|\theta_{t-\tau},X_{t-\tau+1},X_t=x,A_t=a,\theta_{t-1}=z_{t-1})$$

$$\times\Big|\mathbb{P}(X_t=dx|\theta_{t-\tau},X_{t-\tau+1})\pi_{z_{t-1}}(a|x)-\mathbb{P}(\tilde{X}_t=dx|\theta_{t-\tau},X_{t-\tau+1})\pi_{\theta_{t-\tau}}(a|x)\Big|$$

$$=\frac{1}{2}\int_{\mathcal{X}}\sum_{\mathcal{A}}\int_{\mathbb{R}^N}\mathbb{P}(\theta_{t-1}=dz_{t-1}|\theta_{t-\tau},X_{t-\tau+1},X_t=x)\Big|\mathbb{P}(X_t=dx|\theta_{t-\tau},X_{t-\tau+1})\pi_{z_{t-1}}(a|x)$$

$$-\mathbb{P}(\tilde{X}_t=dx|\theta_{t-\tau},X_{t-\tau+1})\pi_{\theta_{t-\tau}}(a|x)\Big| \tag{62}$$

To further bound (62), we play a similar trick as the one in (56). It follows that

$$\frac{1}{2}\int_{\mathcal{X}}\sum_{\mathcal{A}}\int_{\mathbb{R}^N}\mathbb{P}(\theta_{t-1}=dz_{t-1}|\theta_{t-\tau},X_{t-\tau+1},X_t=x)\Big|\mathbb{P}(X_t=dx|\theta_{t-\tau},X_{t-\tau+1})\pi_{z_{t-1}}(a|x)$$

$$-\mathbb{P}(\tilde{X}_t=dx|\theta_{t-\tau},X_{t-\tau+1})\pi_{\theta_{t-\tau}}(a|x)\Big|$$

$$\leq\frac{1}{2}\int_{\mathcal{X}}\sum_{\mathcal{A}}\int_{\mathbb{R}^N}\mathbb{P}(\theta_{t-1}=dz_{t-1}|\theta_{t-\tau},X_{t-\tau+1},X_t=x)\Big(\Big|\mathbb{P}(X_t=dx|\theta_{t-\tau},X_{t-\tau+1})\pi_{z_{t-1}}(a|x)$$

$$-\mathbb{P}(\tilde{X}_t=dx|\theta_{t-\tau},X_{t-\tau+1})\pi_{z_{t-1}}(a|x)\Big|$$

$$+\Big|\mathbb{P}(\tilde{X}_t=dx|\theta_{t-\tau},X_{t-\tau+1})\pi_{z_{t-1}}(a|x)-\mathbb{P}(\tilde{X}_t=dx|\theta_{t-\tau},X_{t-\tau+1})\pi_{\theta_{t-\tau}}(a|x)\Big|\Big)$$

$$=\frac{1}{2}\int_{\mathcal{X}}\Big|\mathbb{P}(X_t=dx|\theta_{t-\tau},X_{t-\tau+1})-\mathbb{P}(\tilde{X}_t=dx|\theta_{t-\tau},X_{t-\tau+1})\Big|$$

$$+\frac{1}{2}\int_{\mathcal{X}}\sum_{\mathcal{A}}\int_{\mathbb{R}^N}\mathbb{P}(\theta_{t-1}=dz_{t-1}|\theta_{t-\tau},X_{t-\tau+1},X_t=x)\mathbb{P}(\tilde{X}_t=dx|\theta_{t-\tau},X_{t-\tau+1})$$

$$\times\Big|\pi_{z_{t-1}}(a|x)-\pi_{\theta_{t-\tau}}(a|x)\Big|$$

$$\leq\frac{1}{2}\int_{\mathcal{X}}\Big|\mathbb{P}(X_t=dx|\theta_{t-\tau},X_{t-\tau+1})-\mathbb{P}(\tilde{X}_t=dx|\theta_{t-\tau},X_{t-\tau+1})\Big|$$

$$+\int_{\mathcal{X}}\int_{\mathbb{R}^N}\mathbb{P}(\theta_{t-1}=dz_{t-1}|\theta_{t-\tau},X_{t-\tau+1},X_t=x)\mathbb{P}(\tilde{X}_t=dx|\theta_{t-\tau},X_{t-\tau+1})$$

$$\times\sup_{x\in\mathcal{X}}\|\pi_{z_{t-1}}(\cdot|x)-\pi_{\theta_{t-\tau}}(\cdot|x)\|_{TV} \tag{63}$$

The first term in (63) is

$$\|\mathbb{P}(X_t\in\cdot|\theta_{t-\tau},X_{t-\tau+1})-\mathbb{P}(\tilde{X}_t\in\cdot|\theta_{t-\tau},X_{t-\tau+1})\|_{TV}. \tag{64}$$

The second term can be bounded similar to (58). More specifically, recalling that

$$\|\theta_t - \theta_{t-\tau}\|_2 \leq \sum_{i=t-\tau}^{t-1} \|\theta_{i+1} - \theta_i\|_2 \leq G \sum_{i=t-\tau}^{t-1} \alpha_i, \tag{65}$$

and by the Lipschitz condition in (2), it follows that for any $x \in \mathcal{X}$ and any $z_{t-1}$ such that $z_{t-1} dz_{t-1}$ has non-zero measure, then

$$\|\pi_{z_{t-1}}(\cdot|x) - \pi_{\theta_{t-\tau}}(\cdot|x)\|_{TV} \leq C|\mathcal{A}| \|z_{t-1} - \theta_{t-\tau}\|_2 \leq C|\mathcal{A}|G \sum_{i=t-\tau}^{t-2} \alpha_i. \tag{66}$$

Thus, (63) is upper bounded by

$$\|\mathbb{P}(X_t \in \cdot|\theta_{t-\tau}, X_{t-\tau+1}) - \mathbb{P}(\tilde{X}_t \in \cdot|\theta_{t-\tau}, X_{t-\tau+1})\|_{TV} + C|\mathcal{A}|G \sum_{i=t-\tau}^{t-2} \alpha_i \tag{67}$$

## C   Proof of Theorem 1

We decompose the error as follows,

$$\mathbb{E}[\|\theta_{t+1} - \theta^*\|_2^2]$$
$$= \mathbb{E}[\|\mathrm{proj}_{2,R}(\theta_t + \alpha_t g_t(\theta_t)) - \mathrm{proj}_{2,R}(\theta^*)\|_2^2]$$
$$\leq \mathbb{E}[\|\theta_t + \alpha_t g_t(\theta_t) - \theta^*\|_2^2]$$
$$= \mathbb{E}[\|\theta_t - \theta^*\|_2^2 + \alpha_t^2 \|g_t(\theta_t)\|_2^2 + 2\alpha_t \langle \theta_t - \theta^*, g_t(\theta_t) \rangle]$$
$$= \mathbb{E}[\|\theta_t - \theta^*\|_2^2 + \alpha_t^2 \|g_t(\theta_t)\|_2^2 + 2\alpha_t \langle \theta_t - \theta^*, \bar{g}(\theta_t) - \bar{g}(\theta^*) \rangle + 2\alpha_t \mathbf{\Lambda}_t(\theta_t)], \tag{68}$$

where the inequality is due to the fact that orthogonal projections onto a convex set are non-expansive, and the last step is due to the fact that $\bar{g}(\theta^*) = 0$. Applying Lemmas 2 and 4 implies that

$$\mathbb{E}[\|\theta_{t+1} - \theta^*\|_2^2] \leq \mathbb{E}[(1 - 2\alpha_t w_s) \|\theta_t - \theta^*\|_2^2 + \alpha_t^2 G^2 + 2\alpha_t \mathbf{\Lambda}_t(\theta_t)], \tag{69}$$

which further implies that

$$w(t+1)\mathbb{E}[\|\theta_{t+1} - \theta^*\|_2^2] \leq \mathbb{E}[tw \|\theta_t - \theta^*\|_2^2 + \frac{1}{2}\alpha_t G^2 + \mathbf{\Lambda}_t(\theta_t)]. \tag{70}$$

Applying (70) recursively and Lemma 7 (with $\tau = \tau_0$) yields that

$$wT\mathbb{E}[\|\theta_T - \theta^*\|_2^2]$$
$$\leq \sum_{t=0}^{T-1} \left( \frac{1}{2}\alpha_t G^2 + \mathbb{E}[\mathbf{\Lambda}_t(\theta_t)] \right)$$
$$\leq \frac{(\log T + 1)G^2}{4w} + \sum_{t=0}^{\tau_0} \mathbb{E}[\mathbf{\Lambda}_t(\theta_t)] + \sum_{t=\tau_0+1}^{T-1} \mathbb{E}[\mathbf{\Lambda}_t(\theta_t)]$$
$$\leq \frac{(\log T + 1)G^2}{4w} + 2G^2(\tau_0 + 1) + \frac{(C|\mathcal{A}|G^3\tau_0^2 + (6 + \lambda C)G^2\tau_0/2)(\log T + 1)}{w} + \frac{2G^2}{\rho w}$$
$$= \frac{G^2(4C|\mathcal{A}|G\tau_0^2 + (12 + 2\lambda C)\tau_0 + 1)(\log T + 1)}{4w} + \frac{2G^2(\tau_0 w + w + \rho^{-1})}{w}, \tag{71}$$

which completes the proof.

## D   Proof of Lemma 1

Consider $A_\theta = \mathbb{E}_\theta[\phi(X, A)(\gamma\phi^T(Y, B)\theta - \phi^T(X, A)\theta)]$, and $b_\theta = \mathbb{E}_\theta[\phi(X, A)r(X, A)]$ as defined in Section 3.2 with respect to $\theta$. It has been shown that for any $\theta \in \mathbb{R}^N$, $A_\theta$ is negative definite [28, 36]. Denote by $w_l$ the largest eigenvalue of $\frac{1}{2}(A_{\theta^*} + A_{\theta^*}^T)$. Recall that the limit point $\theta^*$ satisfies the following relationship $-A_{\theta^*}\theta^* = b_{\theta^*}$ (Theorem 2 [23]). It then follows that $-(\theta^*)^T A_{\theta^*}\theta^* = (\theta^*)^T b_{\theta^*}$, which implies $- w_l \|\theta^*\|_2^2 \leq \|\theta^*\|_2 r_{\max}$. Thus, $\|\theta^*\|_2 \leq -\frac{r_{\max}}{w_l}$, which completes the proof.

# E   Proof of Theorem 2

The proof for Theorem 2 with constant step size is similar to the proof of Theorem 1. In the following, we mainly outline the difference between them.

First, we obtain the following results from Lemma 7 by letting $\tau = t$ if $t \leq \tau_0$, and $\tau = \tau_0$ if $t > \tau_0$:

$$\mathbb{E}[\mathbf{\Lambda}_t(\theta_t)] \leq (6 + \lambda C)G^2 \tau_0 \alpha_0 + 4G^2 \alpha_0/\rho + 2G^3 C|\mathcal{A}|\tau_0^2 \alpha_0, \tag{72}$$

where $\tau_0 = \inf\{t \geq 0 : m\rho^t \leq \alpha_0\}$.

We then decompose the error following similar steps to those in (68), and we obtain that

$$
\begin{aligned}
\mathbb{E}[\, &\|\theta_{t+1} - \theta^*\|_2^2] \\
&\leq \mathbb{E}[\|\theta_t - \theta^*\|_2^2 + \alpha_0^2 \|g_t(\theta_t)\|_2^2 + 2\alpha_t \langle \theta_t - \theta^*, \bar{g}(\theta_t) - \bar{g}(\theta^*) \rangle + 2\alpha_0 \mathbf{\Lambda}_t(\theta_t)] \\
&\leq \mathbb{E}[\|\theta_t - \theta^*\|_2^2 + \alpha_0^2 G^2 - 2\alpha_0 w_s \|\theta_t - \theta^*\|_2^2 + 2\alpha_0 \mathbf{\Lambda}_t(\theta_t)] \\
&= \mathbb{E}[(1 - 2\alpha_0 w_s) \|\theta_t - \theta^*\|_2^2 + \alpha_0^2 G^2 + 2\alpha_0 \mathbf{\Lambda}_t(\theta_t)]
\end{aligned}
\tag{73}
$$

where the last inequality follows from Lemma 4. Applying (73) recursively with (72), we obtain Theorem 2.

# F   Proof of Theorem 3

Before we start the proof of Theorem 3, we first present the following lemma.

**Lemma 8.** *Consider a non-increasing step-size sequence $\alpha_0 \geq \alpha_1 ... \geq \alpha_T$. For any fixed $\tau$, and $tB \leq i \leq (t+1)B - 1$, if $tB \leq \tau$, then we have that*

$$\mathbb{E}[\langle \theta_{tB} - \theta^*, g_i(\theta_i) - \bar{g}(\theta_{tB}) \rangle] \leq 2G^2; \tag{74}$$

*and if $tB > \tau$, then*

$$
\begin{aligned}
\mathbb{E}[&\langle \theta_{tB} - \theta^*, g_i(\theta_i) - \bar{g}(\theta_{tB}) \rangle] \\
&\leq (6 + \lambda C)G^2(\tau + B)\alpha_{tB-\tau} + 4G^2 m\rho^{\tau-1} + C|\mathcal{A}|G^3 \tau^2 \alpha_{tB-\tau}.
\end{aligned}
\tag{75}
$$

*Proof.* For any fixed $\tau$, if $tB \leq \tau$, then we use the following upper bound:

$$\mathbb{E}[\langle \theta_{tB} - \theta^*, g_i(\theta_i) - \bar{g}(\theta_{tB}) \rangle] \leq 4RG \leq 2G^2, \tag{76}$$

and if $tB > \tau$, then for any $tB \leq i \leq (t+1)B - 1$, it follows that

$$
\begin{aligned}
\mathbb{E}[&\langle \theta_{tB} - \theta^*, g_i(\theta_i) - \bar{g}(\theta_{tB}) \rangle] \\
&= \mathbb{E}[\langle \theta_{tB-\tau} - \theta^*, g_i(\theta_{tB-\tau}) - \bar{g}(\theta_{tB-\tau}) \rangle] \\
&\quad + \mathbb{E}[\langle \theta_{tB} - \theta^*, g_i(\theta_i) - \bar{g}(\theta_{tB}) \rangle] - \mathbb{E}[\langle \theta_{tB-\tau} - \theta^*, g_i(\theta_{tB-\tau}) - \bar{g}(\theta_{tB-\tau}) \rangle].
\end{aligned}
\tag{77}
$$

The difference between the second and third terms in (77) can be bounded as follows:

$$
\begin{aligned}
\mathbb{E}[&\langle \theta_{tB} - \theta^*, g_i(\theta_i) - \bar{g}(\theta_{tB}) \rangle] - \mathbb{E}[\langle \theta_{tB-\tau} - \theta^*, g_i(\theta_{tB-\tau}) - \bar{g}(\theta_{tB-\tau}) \rangle] \\
&= \mathbb{E}[\langle \theta_{tB} - \theta^*, g_i(\theta_i) - \bar{g}(\theta_{tB}) \rangle] - \mathbb{E}[\langle \theta_{tB} - \theta^*, g_i(\theta_{tB-\tau}) - \bar{g}(\theta_{tB-\tau}) \rangle] \\
&\quad + \mathbb{E}[\langle \theta_{tB} - \theta^*, g_i(\theta_{tB-\tau}) - \bar{g}(\theta_{tB-\tau}) \rangle] - \mathbb{E}[\langle \theta_{tB-\tau} - \theta^*, g_i(\theta_{tB-\tau}) - \bar{g}(\theta_{tB-\tau}) \rangle] \\
&= \mathbb{E}[\langle \theta_{tB} - \theta^*, g_i(\theta_i) - g_i(\theta_{tB-\tau}) \rangle] - \mathbb{E}[\langle \theta_{tB} - \theta^*, \bar{g}(\theta_{tB}) - \bar{g}(\theta_{tB-\tau}) \rangle] \\
&\quad + \mathbb{E}[\langle \theta_{tB} - \theta_{tB-\tau}, g_i(\theta_{tB-\tau}) - \bar{g}(\theta_{tB-\tau}) \rangle] \\
&\overset{(a)}{\leq} 2R(1+\gamma) \|\theta_i - \theta_{tB-\tau}\|_2 + 2R(1+\gamma+\lambda C) \|\theta_{tB} - \theta_{tB-\tau}\|_2 + 2G \|\theta_{tB} - \theta_{tB-\tau}\|_2 \\
&\leq 2G^2 \sum_{j=tB-\tau}^{i-1} \alpha_j + (4 + \lambda C)G^2 \sum_{j=tB-\tau}^{tB-1} \alpha_j \\
&\overset{(b)}{\leq} (6 + \lambda C)G^2(\tau + B)\alpha_{tB-\tau}.
\end{aligned}
\tag{78}
$$

Here the step (a) in the above equation follows similarly to those in (29) and (30), and (b) is due to the fact that the sequence $\alpha_j$'s is non-increasing.

Next, we consider the first term in (77), which can be bounded using similar steps as in the proof of Lemma 7. In particular, let $\tau = n_1 B$, for some positive integer $n_1$. We then design an auxiliary Markov chain following a fixed policy given by $\Gamma(\phi^T \theta_{tB-\tau})$. It can be shown that

$$\mathbb{E}[\langle \theta_{tB-\tau} - \theta^*, g_i(\theta_{tB-\tau}) - \bar{g}(\theta_{tB-\tau})\rangle]$$
$$\leq 4G^2 m \rho^{\tau-1} + 2C|\mathcal{A}|G^3 \sum_{j=tB-\tau}^{tB-1} \sum_{i=tB-\tau}^{j} \alpha_i$$
$$\leq 4G^2 m \rho^{\tau-1} + C|\mathcal{A}|G^3 \tau^2 \alpha_{tB-\tau}. \tag{79}$$

This completes the proof. □

Now we are ready to prove Theorem 3. Following similar steps to those in the proof of Theorem 1, the error at time $B(t+1)$ can be decomposed as follows:

$$\mathbb{E}[\left\|\theta_{(t+1)B} - \theta^*\right\|_2^2]$$
$$\leq \mathbb{E}\left[\left\|\theta_{tB} + \sum_{i=tB}^{(t+1)B-1} \alpha_i g_i(\theta_i) - \theta^*\right\|_2^2\right]$$
$$= \mathbb{E}\left[\|\theta_{tB} - \theta^*\|_2^2 + 2 \sum_{i=tB}^{(t+1)B-1} \alpha_i \langle \theta_{tB} - \theta^*, g_i(\theta_i)\rangle + \left\|\sum_{i=tB}^{(t+1)B-1} \alpha_i g_i(\theta_i)\right\|_2^2\right]. \tag{80}$$

The third term in (80) can be upper bounded as follows:

$$\left\|\sum_{i=tB}^{(t+1)B-1} \alpha_i g_i(\theta_i)\right\|_2^2 \leq \left(\sum_{i=tB}^{(t+1)B-1} \alpha_i G\right)^2 \leq B^2 G^2 \alpha_{tB}^2. \tag{81}$$

We then consider the second term in (80). It can be shown that

$$\mathbb{E}[\langle \theta_{tB} - \theta^*, g_i(\theta_i)\rangle]$$
$$= \mathbb{E}[\langle \theta_{tB} - \theta^*, \bar{g}(\theta_{tB}) - \bar{g}(\theta^*)\rangle] + \mathbb{E}[\langle \theta_{tB} - \theta^*, g_i(\theta_i) - \bar{g}(\theta_{tB})\rangle], \tag{82}$$

which is due to the fact that $\bar{g}(\theta^*) = 0$. The first term in the above equation can be upper bounded using Lemma 4, i.e.,

$$\mathbb{E}[\langle \theta_{tB} - \theta^*, \bar{g}(\theta_{tB}) - \bar{g}(\theta^*)\rangle] \leq -w_s \mathbb{E}\|\theta_{tB} - \theta^*\|_2^2. \tag{83}$$

The second term in (82) can be bounded using Lemma 8.

It then follows that

$$\mathbb{E}[\left\|\theta_{(t+1)B} - \theta^*\right\|_2^2] \leq \mathbb{E}[(1 - 2w_s B \alpha_{(t+1)B}) \|\theta_{tB} - \theta^*\|_2^2] + B^2 G^2 \alpha_{tB}^2$$
$$+ 2 \sum_{i=tB}^{(t+1)B-1} \alpha_i \mathbb{E}[\langle \theta_{tB} - \theta^*, g_i(\theta_i) - \bar{g}(\theta_{tB})\rangle]. \tag{84}$$

For the case with diminishing step size, i.e., $\alpha_t = \frac{1}{2tw}$, it follows that

$$(t+1)\mathbb{E}[\left\|\theta_{(t+1)B} - \theta^*\right\|_2^2] \leq t\mathbb{E}[\|\theta_{tB} - \theta^*\|_2^2] + B^2 G^2 \alpha_{tB}^2(t+1)$$
$$+ 2(t+1) \sum_{i=tB}^{(t+1)B-1} \alpha_i \mathbb{E}[\langle \theta_{tB} - \theta^*, g_i(\theta_i) - \bar{g}(\theta_{tB})\rangle]. \tag{85}$$

Applying the above inequality recursively, we obtain that for any $\tau = n_1 B > 0$ where $n_1$ is some positive integer,

$$T\mathbb{E}[\|\theta_{TB} - \theta^*\|_2^2]$$

$$\leq \sum_{t=0}^{T-1}\left(B^2 G^2 \alpha_{tB}^2(t+1) + 2(t+1)\sum_{i=tB}^{(t+1)B-1}\alpha_i\mathbb{E}[\langle\theta_{tB} - \theta^*, g_i(\theta_i) - \bar{g}(\theta_{tB})\rangle]\right)$$

$$\leq \frac{G^2}{2w^2}(\log T + 2) + \frac{4G^2(\tau + B)}{wB} + \frac{(6+\lambda C)G^2(\tau + B) + C|\mathcal{A}|G^3\tau^2}{Bw^2}(\log T + 1)$$

$$+ \frac{8G^2 m\rho^{\tau-1}T}{w}, \tag{86}$$

where we let $\alpha_0 = \frac{1}{\sqrt{2}Bw}$. If we further let $\tau = \tau_0 = \inf\{nB : m\rho^{nB} \leq \alpha_{TB}\}$, then it follows that

$$\mathbb{E}[\|\theta_{TB} - \theta^*\|_2^2]$$

$$\leq \Big(4G^2(\tau_0 + B)w + (\log T + 1)((6+\lambda C)G^2\tau_0 + (6.5+\lambda C)G^2 B$$

$$+ C|\mathcal{A}|G^3\tau_0^2) + 4G^2/\rho + 0.5BG^2\Big)/\Big(w^2 BT\Big). \tag{87}$$

For the case with constant step size, i.e., $\alpha_t = \alpha_0 < \frac{1}{2w_s B}$, we first show that

$$\mathbb{E}[\langle\theta_{tB} - \theta^*, g_i(\theta_i) - \bar{g}(\theta_{tB})\rangle]$$

$$\leq (6+\lambda C)G^2(\tau_0 + B)\alpha_0 + 4G^2 m\rho^{\tau_0-1} + C|\mathcal{A}|G^3\tau_0^2\alpha_0, \tag{88}$$

by letting $\tau = tB$ if $tB \leq \tau_0$, and $\tau = \tau_0$ if $tB > \tau_0$ in Lemma 8. Then, applying (84) recursively, we obtain that

$$\mathbb{E}[\|\theta_{TB} - \theta^*\|_2^2]$$

$$\leq e^{-2w_s B\alpha_0 T}\|\theta_0 - \theta^*\|_2^2 + \frac{\alpha_0(BG^2 + 2(6+\lambda C)G^2(\tau_0 + B) + 8G^2/\rho + 2|\mathcal{A}|G^3\tau_0^2)}{2w_s}. \tag{89}$$