[Reviews · NeurIPS 2019]

Reviewer 1



This paper deals with an important problem in theoretical reinforcement learning (RL), that is, finite-time analysis of on-policy RL algorithms such as SARSA. If the analysis techniques, as well as proofs, were correct and concrete, this work may have a broad impact on analyzing related stochastic approximation/RL algorithms. Although important and interesting, the present submission contains several major concerns, that have limited the contributions and even brought into question the practical usefulness of the reported theoretical results. These concerns are listed as follows. 1. To facilitate analysis, a number of the assumptions adopted in this work are strong and impractical. For example, i) The projection step in (1) is impractical. In addition, if one notices, later in the proof of Lemma 2 (see eq. (27) in the Appendix), another assumption requires the norm of feature vectors to be less than 1. Making these two assumptions simultaneously, does not necessarily guarantee that the true Q-function can be approximated by the inner product of $\theta$ and $\phi$. ii) Later, it is claimed that without the projection, the method is still useful, and one just needs to set $R$ to possibly a very large value; however, it can be seen from the right-hand side of equations (3), (4), and (11), that the second and third power of $R$ shows up through $G$, and thus the bounds will possibly be very loose. iii) Besides the projection step to bound the gradient norm, another related concern was also raised by [Chen et. al. in Finite-Time Analysis of Q-Learning with Linear Function Approximation, arXiv:905.11425v1, May 2019]. It is claimed in [Chen etal'19] that the results (Theorem 1) of [F. S. Melo et. al. Analysis of reinforcement learning with function approximation, ICML2008], that this work builds on, cannot be verified. This concern further weakens the contributions of this work. iv) Assumption 2 requires $C$ to be small enough to guarantee a negative eigenvalue. This assumption is vague, in the sense that there is no clear characterization of $w_s$, and in practice, how one can guarantee a "feasible" or "meaningful" policy for which it is possible to have such a small $C$. 2. It is also recommended to provide some numerical tests to verify the theoretical results.

Reviewer 2



This paper builds on recent work on finite-time analysis of linear TD learning and provides finite-time results for linear Sarsa. The key insight is constructing a Markov Decision Process for each policy obtained during the optimization. The contribution is incremental yet significant. Non-asymptotic behavior of Sarsa with Markovian samples is still an open problem and these results will be interesting to the community. The paper is well-organized and well-written. I have pointed out a few small suggestions to make the proof easier to follow. Minor comments: - Write the proof for Lemma 5 explicitly. - Remind the reader that Eq (50) follows from (49) because of the relationship between \alpha_t and \w_s. - The environment's transition dynamics is denoted by P while P_\theta shows the invariant measure (rather than the transition dynamics) induced by \theta. This notation is slightly confusing. I suggest using both P and P_\theta for transition dynamics (the latter being induced by \theta). Questions: - How realistic is assumption that for any \theta the induced Markov chain is ergodic (right before Assumption 1)? Is it guaranteed for any special class of environments or representations? - Does this result bear implications for framing an RL problem or designing features? ---------------------------------------------------------------------- Update: The projection step in the algorithm is problematic because the analyzed algorithm is different from Sarsa and the projection radius can have a large impact on the bounds. I was initially not concerned with the projection step. I had accepted that it was a limitation of a recent work [4] and that eliminating it was not in the scope of the submitted paper. After discussions with the other reviewers, I became aware that a different line of analysis had already avoided this extra step and provided finite-time bounds for TD with standard and practical assumptions [33]. I lower my score as it is not clear why the submitted paper did not pursue the direction that studies unmodified RL algorithms.

Reviewer 3



The problem studied here is important since there is no previous finite-sample result for SARSA/policy-improvement without an converged value iteration inside of it, and it is helpful for us to understand the learning dynamics of policy improvement type of algorithm. This type of finite-sample analysis for SARSA is technically interesting to me due to the two challenge mentioned above. The method in this paper is novel to my knowledge. The only concern I have is that the analysis is limited in linear function approximation case. In general I am happy to see this paper get accepted. === Update === After read the author's rebuttal and discussion from other reviewers, I have some updates: 1. This paper studies the finite sample guarantees on time-varying markov chain, caused by the updating behavior policy during collecting samples. The key insight to deal with this difficulty is novel and clear. 2. As other reviewers pointed out, the proofs seems heavily rely the projection step. The projection radius is used to bound the update term g_t. However some recent work on finite sample analysis of SARSA/Q-learning seems to work without this assumption [33,37]. Given the recent advances, the assumption here seems too strong and also not really necessary. 3. Another concern is about the \theta^* mentioned in the paper as "limit point of SARSA". It is not clear if SARSA with/without the projection have the same limit point or not , which need to clarified. (Even the limit point it self is in the radius, the update dynamics could change.) 4. It is a little bit hard for me to understand how strong assumption 2 is and verified that (even conceptually) with some instances. This assumption is not very intuitive since it depends on the global information of MDPs. Overall, the novel analysis here is indeed interesting and valuable. So I tend to vote for a weakly accept, though I still keep my concerns as listed in 2,3,4 above.

Reviewer 4



I. Contributions of the Paper: In this paper the authors gave the first finite-sample analysis of two convergent SARSA(0) variants, for a class of MDPs that have continuous state space and finite action space and satisfy a uniform geometric ergodicity condition (Assumption 1, Section 3.1). The two SARSA(0) variants are based on the convergent form of approximate policy iteration proposed by Perkins and Precup [27]; the asymptotic convergence analysis of one of the algorithms was given by Melo et al. [21]. To distinguish these convergent SARSA(0) variants from the original non-convergent SARSA algorithm, I shall refer to them as C-SARSA in what follows. The main difficulty and complexity in finite-sample analysis of C-SARSA arise from the time-varying Markov chains underlying the C-SARSA iterates. In order to derive finite-sample bounds, the authors introduced a new technique that combines Mitrophanov's perturbation bound [23] for uniformly ergodic Markov chains with careful and novel ways of decomposing the error terms in the analysis of C-SARSA. Their analysis is very interesting and requires a lot of efforts. It is not a straightforward application of Mitrophanov's bound. Some of the other arguments that the authors introduced in the proofs are equally important, in order to make full use of Mitrophanov's result and derive effective finite-sample bounds. These proof techniques are solid and important contributions, and I agree with the authors that besides C-SARSA, they will also be useful tools for analyzing other similar types of algorithms that involve time-varying Markov chains. II. Technical Correctness: I think some corrections are needed in the constants involved in the finite-sample bounds due to an oversight in bounding one term in the proof. Also, some conditions in Sections 2 and 3 need to be stated more rigorously. (See the detailed comments below.) The overall conclusions are not affected, however. The proofs are mostly well-done and presented with great clarity. III. Shortcoming of the paper: A big shortcoming is that the authors ignored the important question whether the fixed points of C-SARSA are good policies and did not discuss at all the limitations of the assumptions involved. The most critical assumption is Assumption 2 (Section 3.2), which is essentially the same kind of condition as in Perkins and Precup [27]. It is a stringent smoothness condition on the "policy improvement operator," which C-SARSA needs in order to converge. It's plain to see that such a strong smoothness condition could make the fixed points of C-SARSA useless -- I'll give a simple one-state MDP example to illustrate this point (see Part C below). The authors devoted one section (Section 5) to mainly justify such smoothness conditions based on the convergence guarantee they bring, without mentioning at all their troublesome consequences. If convergence is the only thing that matters, then we could just let the Lipschitz constant be zero and get C-SARSA to converge in one iteration. Does this make sense? I expect that C-SARSA would have trouble finding useful solutions, especially for large discount factors. If, contrary to what I'd expect, the fixed points of C-SARSA are not uniformly bad, then the authors should give one or two examples to convince readers that this algorithm is able to learn good policies in some cases. Otherwise, I think in the future the authors should consider using an algorithm with better performance guarantees to demonstrate their new techniques for finite-sample analysis. ---------------------------------- More detailed comments below -------------------- A. Detailed comments on technical issues: 1. Proof of Lemma 6, line 450, p. 13: The term $\| \bar{g}(\theta_1) - \bar{g}(\theta_2) \|$ cannot be bounded by $(1 + \gamma) \| \theta_1 - \theta_2 \|$ as the authors claimed, since it involves the invariant distributions of the policies corresponding to $\theta_1$ and $\theta_2$. Calculations similar to (20)-(22) in the proof of Lemma 4 need to performed, so the constant $\lambda_1 + \lambda_2$ will also appear in the bound of this term. Then the constants in the finite-sample bounds in Theorems 1-3 need to be corrected accordingly. 2. Proof of Theorem 3, Eq. (60), p. 17: From the first equality in (60), it seems that the authors are applying projection only at the end of every $B$ updates of the $\theta$'s with the same policy, instead of applying the projection at each update as stated in Algorithm 2 (p. 7). The proof and/or the algorithm need to be modified to make them consistent with each other. 3. Line 475-479, p. 14: Several total-variation terms have strange expressions that involve two different random state variables as conditioning variables: the last two terms in (39) and the left-hand sides of (40) and (41). I think what the authors meant are these two terms instead: the conditional expectation (over $\theta_{t-1}$) of $\max_{x \in X} \| \pi_{\theta_{t-1}}(\cdot \mid x ) - \pi_{\theta_{t-\tau}}(\cdot \mid x) \|_{TV}$ given $(\theta_{t-\tau}, X_{t-\tau +1})$ and the conditional expectation (over $\theta_t$) of $\max_{x \in X} \| \pi_{\theta_t}(\cdot \mid x ) - \pi_{\theta_{t-\tau}}(\cdot \mid x) \|_{TV}$ given $(\theta_{t-\tau}, X_{t-\tau +1})$. 4. Section 3.2, p. 5: As the matrix $A_{\theta^*}$ is asymmetric, the number $w_l$ appearing in Lemma 1 is not the eigenvalue of $A_{\theta^*}$. The statement of Assumption 2 has a similar problem with the number $-w_s$. Corrections are needed. 5. Theorem 1, p. 5: It would be better to use a different number $w \leq w_s$ in defining the diminishing stepsize sequence and in deriving/expressing the bound (3), because $w_s$ is unknown in practice and one can't take stepsize precisely equal to $1/(2 w_s (t+1))$. The same remark for the diminishing stepsize case in Theorem 3, p. 7. 6. About conditions involved: (i) The conditions ensuring the existence and uniqueness of a fixed point $\theta^*$ need to be stated in Section 3. (ii) Both the existence of the fixed point $\theta^*$ and the value of $A_{\theta^*}$ depend on the operator $\Gamma$ and hence also on the Lipschitz constant $C$. But Assumption 2 (p. 5) sounds like $C$ can be chosen independently. This needs to be clarified. (iii) Section 2.2, line 145, p. 4: The statement of the condition that the functions $\phi_i$, $i \leq N$ are "linearly independent" is too loose and imprecise, as the state space here is infinite. If the matrices $A_\theta$ need to be invertible for all $\theta$ (which I think the authors would need as part of the conditions for the existence of a fixed point $\theta^*$), then the functions $\phi_i, i \leq N$ need to be linearly independent in the Hilbert space $L^2(X \times A, \mu_\theta)$ for EVERY $\theta$. (Here $\mu_\theta$ is the invariant probability measure induced by $\pi_\theta$ on the state-action space, and for the space $L^2(X \times A, \mu_\theta)$, two measurable functions on $X \times \A$ are treated as equivalent if they are identical except on a set of $\mu_\theta$-measure zero.) (iv) Line 171, p. 4: "Assumption 1 holds for irreducible and aperiodic Markov chains" -- This statement is loose/incorrect for continuous state-space Markov chains. (v) Line 123, p. 3: The compactness condition on the state space does not seem to be used anywhere in the paper. --------------------------------------- B. Minor presentation-related issues and typos: Throughout the paper the authors frequently referred to $g_t$ as "gradient". But it is not a gradient. This is rather confusing. Line 130, p. 3: "stationary Markov policy" should be "stationary policy" instead. Line 136, p. 3: $V^*$ should be $V^*(x)$. Line 137, p. 3: several typos in this line. Line 157, p. 4: $\gamma$ is missing in the definition of $\Delta_t$. p. 4: Since $P$ denotes a transition kernel, it would be better to use a different symbol instead of $P_\theta$ to denote the invariant probability measure induced by $\pi_\theta$. Line 189, p. 5: "$\theta_{TB}$" should be $\theta_T$ instead. Line 473-474, p. 14: It's confusing to write (37) is this way, omitting the expectation over $\theta_t$ and $\theta_{t-1}$. Eq. (42), p. 14: The notation $P(X_{t-1} = x ... )$ is not right here. It may be better to write $P( d x_{t-1} ...)$ instead. p. 15: The term $(1 - 2 \alpha_t w_s)$ in (49) and $t w_s$ in (50) should be moved inside the expectation and before $\|\theta_t - \theta^*\|_2^2$ of those two equations, respectively. A similar typo occurs in Eq. (53), p. 16. --------------------------------- C. An example to show the problematic nature of C-SARSA: Consider an MDP with one state and two actions $a$ and $b$. The reward of action $b$ is always zero. For action $a$, there are two possibilities: it either incurs reward $1$ or $-1$, so it is either a good action or a bad one. But we don't know which case we're in. Let's take the look-up table representation, so $\theta$ is just the pair of Q-values $(Q(a), Q(b))$. We let the "policy improvement operator" $\Gamma$ be such that if $\theta = (1/(1-\gamma), 0)$, $\pi_\theta(a)$ is almost $1$. This is reasonable, since $1/(1-\gamma)$ is the maximal amount of reward one can attain in this MDP. If action $a$ would have this much advantage over action $b$, it is reasonable for us to definitely favor action $a$. How large must the probability of taking action $a$ be according to $\Gamma$, if the reward of $a$ turns out to be $-1$? To satisfy the smoothness condition, Assumption 2, of C-SARSA, the Lipschitz constant $C$ needs to satisfy $C \lambda \leq 1 - \gamma$ at least. Since $\lambda \geq 4 R$ and $R \geq 1/(1 - \gamma)$ in this case, we must have $C \leq (1 - \gamma)^2/4$. If the reward of $a$ is $-1$, the policy that takes action $a$ has Q-values $(-1/(1-\gamma), -\gamma/(1-\gamma))$. Then using the bound on $C$, we get that at this pair of Q-values, $\Gamma$ still has to give at least probability $p$ to action $a$, where $1 - p \leq 0.56 (1 - \gamma)$. If the discount factor $\gamma = 0.9$, then $p \geq 0.94$. Thus, despite that we started out reasonably to let $\Gamma$ output high probability for taking action $a$ if $\theta = (10, 0)$, we ended up with the non-sensible requirement due to the smoothness condition of C-SARSA that we must also take action $a$ with almost equally high probability if the reward for $a$ turns out to be $-1$. In the above $\Gamma$ is defined point by point. It's also straightforward to let $\Gamma$ be any popular type of operator used in practice and check how smooth it should be in order to satisfy the required smoothness condition. For example, let $\Gamma(Q(a), Q(b))$ be the vector $(e^{\beta Q(a)}, e^{\beta Q(b)})$ normalized by the sum $e^{\beta Q(a)} + e^{\beta Q(b)}$, where $\beta > 0$ is a parameter. Then, to satisfy $C \leq (1- \gamma)^2/4$, one can verify that the parameter $\beta$ must be no greater than $(1 - \gamma)^2/\sqrt{2}$. For $\gamma = 0.9$, the result is that for whatever policy $\pi$ in this one-state MDP, $\Gamma$ maps $Q_\pi$ to a policy that is almost indistinguishable from the randomized policy that takes actions $a$ and $b$ with equal probability.

[Author Response · NeurIPS 2019]

**Q1.** *The projection step is impractical. In addition, if one notices, later in the proof of Lemma 2 (see eq.*
*(27) in the Appendix), another assumption requires the norm of feature vectors to be less than 1. Making these two*
*assumptions simultaneously, does not necessarily guarantee that the true Q-function can be approximated by $\phi^T \theta$.*
**Response:** As we clarify in our answer to Q2 below, projection is not needed for practically implementing SARSA,
and convergence still holds. Even if it is used, to guarantee approximation of $Q$-function by $\phi^T \theta^*$, as stated in our
Theorem 1, we require that the optimal $\theta^*$ is within the projection radius. Such a projection radius can be obtained in
practice by using our Lemma 1 and designing an online estimator of $w_l$.

**Q2.** *Later, it is claimed that without the projection, the method is still useful, and one just needs to set $R$ to possibly a*
*very large value; however, it can be seen from the right-hand side of equations (3), (4), and (11), that the second and*
*third power of $R$ shows up through $G$, and thus the bounds will possibly be very loose.*
**Response:** We clarify that projection is NOT necessary to implement SARSA and does not affect its convergence based
on existing literature. (Our statement "set $R$ to possibly a very large value" was not accurate.) [Gorden 2001] has
shown that SARSA converges to a bounded region. [Melo et. al. 2008] and [Perkins and Precup 2003] further showed
that SARSA with a Lipschitz continuous policy improvement operator converges even without projection. Thus $\theta_t$
generated by SARSA is already bounded by itself, and projection is not necessary. Thus, the non-asymptotic bound for
SARSA without projection is the same as if we use projection with $R = \max_t \|\theta_t\|_2$ (only for the purpose of analysis).
Moreover, $R$ is determined by the nature of algorithm. If it happens to be large, then the error should be large by nature,
which doesn't mean the bound is loose.

**Q3.** *Another concern was also raised by [Chen et. al. 2019]. It is claimed in [Chen et. al. 2019] that Thm 1 of [Melo et.*
*al. 2008], that this work builds on, cannot be verified. This concern further weakens the contributions of this work.*
**Response:** We clarify that this paper studies SARSA, **not** Q-learning. Our proof for SARSA does not have the issue
pointed out in [Chen et. al. 2019] for Q-learning. We are aware of the fact that one step in the proof of Thm 1 in [ Melo
et. al. 2008] cannot be verified for Q-learning, which might be due to greedy policy and off-policy training taken in
Q-learning. Such an issue does not affect SARSA as an on-policy algorithm.

**Q4.** *Assumption 2 requires $C$ small enough to guarantee a negative eigenvalue. No clear characterization of $w_s$. In*
*practice, how one can guarantee a "feasible" or "meaningful" policy for which it is possible to have such a small $C$.*
**Response:** In practice, one can numerically tune the parameter to find a "feasible" policy improvement operator, e.g., $\beta$
in softmax (which corresponds to a $C$). Assumption 2 (for guaranteeing convergence) can be empirically checked to
provide guidance for parameter tuning.

**Q5.** *It is also recommended to provide some numerical tests to verify the theoretical results.*
**Response:** We have run numerical results and will include them in the revision.

 **Q6.** *Write the proof for Lemma 5 explicitly.* **Response:** Done.

**Q7.** *Remind that Eq (50) follows from (49) because of the relationship between $\alpha_t$ and $w_s$.* **Response:** Done.

**Q8.** *How realistic is assumption that for any $\theta$ the induced Markov chain is ergodic (right before Assumption 1)? Is it*
*guaranteed for any special class of environments or representations? Discussion on the plausibility of assumptions.*
**Response:** First note that a Markov chain is uniformly ergodic if it is irreducible (i.e., possibly get to any state from
any state) and aperiodic [Levin & Peres 2017]. Now consider an environment for which there exists a policy that
can map any state to any state with nonzero probability (i.e., irreducibility holds) and can get back to the same state
aperiodically. (Note that such environments are commonly encoutered in practice.) Then for any $\theta$, as long as the policy
improvement operator explores (i.e., with non-zero probability to take any action at any state), the induced Markov
chain remains to be irreducible and aperiodic, and is hence ergodic. Therefore, such an assumption is realistic, and is
guaranteed for aforementioned environments. Please see response to Q1 and Q4 for other assumptions.

**Q9.** *Does this result bear implications for framing a RL problem or designing features? Highlight any insights that*
*follow from these results.*
**Response:** First, our result characterizes sample complexity of SARSA for both constant and diminishing step sizes,
which is useful for choosing learning rate to design fast RL algorithms. Second, our result indicates that the faster the
underlying Markov process mixes, the faster SARSA converges. This motivates the design of tricks (e.g., experience
replay for TD and Q-learning [Wang et. al. arXiv:1809.08926 2017]) to improve the mixing property of Markov process.

 **Q10.** *The analysis is limited in linear function approximation case.*
**Response:** One great advantage of linear function approximation is that it is very easy to implement. Even for this
case, there is no existing finite sample analysis in the literature, and this paper accomplishes this step (which is already
technically nontrivial). It is of great interest to further explore more advanced function spaces, such as deep neural
networks. Looking forward, this work serves as a first step towards understanding the more complicated case.

[Meta-Review · NeurIPS 2019]

Because the initial reviews were mixed, I obtained an additional review from an expert in the area of this paper. This 4th review came back clearly positive, but in the mean time one of the positive reviewers changed to negative (and later one of the negatives turned to positive). Then we had a lot of discussion, but the reviewers never did agree on how best to view this paper. In fact, they seemed to talk past each other, and in the end we had two positive and two negative reviews. As the area chair, reading the reviews and listening to the discussion, I found the 4th, very-positive review to be the most compelling. This review contended that the authors make "solid and important contributions" to the theory of reinforcement learning, in particular, to the finite-time analysis of Sarsa. The review notes that the “main difficulty and complexity” in this topic is dealing with the time-varying nature of the Markov chains. The solid and important contributions of this paper are to apply and significantly extend the prior basic theoretical work on time-varying Markov chains by Mitrophanov. The negative reviewers did not contend this view.